# Neural mechanisms of credit assignment for delayed outcomes during contingent learning

**Phillip P Witkowski**[1,2,3†], **Lindsay JH Rondot**[1,2*†], **Zeb Kurth-Nelson**[4], **Mona M Garvert**[5], **Raymond J Dolan**[4,6], **Timothy EJ Behrens**[6,7,8†], **Erie Boorman**[1,2*†]

[1]Center for Mind and Brain, University of California Davis, Davis, United States; [2]Department of Psychology, University of California Davis, Davis, United States; [3]National Institute on Drug Abuse Intramural Research Program, National Institutes of Health, Baltimore, United States; [4]Max Planck University College London Centre for Computational Psychiatry and Ageing Research, University College London, London, United Kingdom; [5]Faculty of Human Sciences, Julius-Maximilians-Universität Würzburg, Würzburg, Germany; [6]Wellcome Centre for Human Neuroimaging, University College London, London, United Kingdom; [7]Wellcome Centre for Integrative Neuroimaging, University of Oxford, John Radcliffe Hospital, Oxford, United Kingdom; [8]Sainsbury Wellcome Centre for Neural Circuits and Behaviour, University College London, London, United Kingdom

*For correspondence:
ljrondot@ucdavis.edu (LJHR);
edboorman@ucdavis.edu (EB)

†These authors contributed equally to this work

## eLife Assessment

This study provides **important** findings that during credit assignment, the lateral orbitofrontal cortex (lOFC) and hippocampus (HC) encode causal choice representations, while the frontopolar cortex (FPl) mediates HC -lOFC interactions when the causality needs to be maintained over longer distractions. This research offers **compelling** evidence and employs sophisticated multivariate pattern analysis. However, while the task design captures the delayed component, it lacks the full complexity and ambiguity of the credit assignment process observed in real-world scenarios. Moreover, the data indicated that other frontal regions beyond just lOFC were involved in delayed credit assignment. This work will be of interest to cognitive and computational neuroscientists who work on value-based decision-making and fronto-hippocampal circuits.

**Abstract** Adaptive behavior in complex environments critically relies on the ability to appropriately link specific choices or actions to their outcomes. However, the neural mechanisms that support the ability to credit only those past choices believed to have caused the observed outcomes remain unclear. Here, we leverage multivariate pattern analyses of functional magnetic resonance imaging (fMRI) data and an adaptive learning task to shed light on the underlying neural mechanisms of such specific credit assignment. We find that the lateral orbitofrontal cortex (lOFC) and hippocampus (HC) code for the causal choice identity when credit needs to be assigned for choices that are separated from outcomes by a long delay, even when this delayed transition is punctuated by interim decisions. Further, we show when interim decisions must be made, learning is additionally supported by lateral frontopolar cortex (lFPC). Our results indicate that lFPC holds previous causal choices in a 'pending' state until a relevant outcome is observed, and the fidelity of these representations predicts the fidelity of subsequent causal choice representations in lOFC and HC during credit assignment. Together, these results highlight the importance of the timely reinstatement of specific causes in lOFC and HC in learning choice-outcome

relationships when delays and choices intervene, a critical component of real-world learning and decision making.

## Introduction

Humans and animals have a remarkable ability to navigate complex environments and infer the likely state of the world from observed phenomena. Such adaptive behavior requires the ability to learn about causal relationships between one's choices and subsequent outcomes. A key challenge for learning systems in the brain arises when a task involves temporal delays between choices and their outcomes. Cooking is one such task in which many decisions may be made about how to adjust the flavor profile of a dish, but the resultant outcomes of these choices typically will not be evaluated until sitting down to eat. Moreover, cooking often requires juggling multiple sub-tasks simultaneously, meaning that interim decisions need to be performed in between adding an ingredient and observing its effect on the dish's flavor. In such cases, discerning the causal relationship between a particular choice and possible outcomes is nontrivial. While this ability to link choices and outcomes is critical to success in real-world tasks, little is known about *how* these links are forged at the neural level.

A large body of pioneering work focusing on the role of the lateral orbitofrontal cortex (lOFC) has highlighted the importance of this region in contingent learning (*Gardner and Schoenbaum, 2021*; *Murray and Rudebeck, 2018*; *Rushworth et al., 2011*). Recent studies in multiple species have emphasized a special role for lOFC in leveraging task knowledge for credit assignment, linking specific reinforcement outcomes to specific past choices (*Boorman et al., 2013*; *Jocham et al., 2016*; *Lamba et al., 2023*; *Stalnaker et al., 2015*; *Sutton and Barto, 2014*; *Walton et al., 2010*). In one key study, lesions to the macaque lOFC, impaired the ability of animals to use a model of the task structure in order to track the contingency between specific choices and outcomes they caused, with credit erroneously spreading to non-causal choices (*Walton et al., 2010*). These results suggest that lOFC is required for using a model of the task structure to form, or update, an association between specific choices and outcomes. Such findings were subsequently replicated and extended in both rats and humans (*Costa et al., 2023*; *Noonan et al., 2017*). Other studies in humans have shown that outcome-related blood oxygen-level-dependent (BOLD) activity in lOFC is specific to contingent, but not non-contingent, reward observations (*Jocham et al., 2016*), and the magnitude of activity reflects the degree to which credit for an outcome is assigned (*Boorman et al., 2013*; *Boorman et al., 2016*). Collectively, these findings suggest that computations within the lOFC are critical to credit assignment; however, little is known about the mechanisms by which the lOFC supports assigning credit for outcomes to specific causes.

One possible mechanism by which the brain assigns credit when reinforcement is delayed is by reinstating a representation of the causal choice at the time of feedback. In principle, this could enable the choice representation to be associated with the online encoding of the outcome, potentially via changes in synaptic plasticity between co-active neuronal ensembles. Such coding of past choices specifically at the time of feedback has been identified in macaque lOFC neuronal ensembles, albeit in the absence of any task requirement for contingent learning (*Tsujimoto et al., 2009*). Likewise, altered dopaminergic prediction error responses in lOFC-lesioned rats were elegantly accounted for by a computational model that incorporates a loss of internal representations of an outcome-linked choice, leading to misattributing value across states (*Takahashi et al., 2011*). Information about previous choices is also found in regions to which the lOFC shares reciprocal connectivity, particularly the hippocampus (HC) (*Barbas and Blatt, 1995*; *Wikenheiser and Schoenbaum, 2016*). A largely separate literature focusing on HC has shown reinstatement of neural activity patterns previously elicited by a stimulus both at the time of choice and reward in sensory pre-conditioning paradigms (*Barron et al., 2020*; *Kurth-Nelson et al., 2015*; *Wimmer and Shohamy, 2012*), and likewise during associative inference and integration (*Koster et al., 2018*; *Park et al., 2020*; *Zeithamova et al., 2012*). Such hippocampal reinstatement of stimulus identity representations might be expected to support lOFC coding of relevant past choices for credit assignment, particularly following lengthier delays (*Foerde and Shohamy, 2011*; *Shohamy et al., 2009*; *Wang et al., 2020*).

In complex tasks where subsequent decisions intervene on the transitions between choices and resultant outcomes, the neural regions supporting credit assignment may extend to encompass regions that also support maintaining information about causal choices pending their resultant outcome. This

would allow learning systems to precisely assign credit to causal choices by bridging over interim decisions that may otherwise be inappropriately linked to the observed outcome. A key region for maintaining such 'pending' information is the lateral frontal pole (lFPC), which has been implicated in maintaining information about prospective actions or cognitive processes that must be delayed and performed in the future (*Burgess et al., 2007*; *Burgess et al., 2011*; *Burgess et al., 2022*). Other research has shown that lFPC activity reflects the reliability of pending alternative task sets (*Donoso et al., 2014*; *Koechlin et al., 2003*; *Koechlin and Hyafil, 2007*), and that it tracks evidence favoring adapting behavior to specific counterfactual alternatives, and directed exploratory choices, in the future (*Badre et al., 2012*; *Boorman et al., 2009*; *Boorman et al., 2011*; *Zajkowski et al., 2017*). On this basis, we hypothesized that the lFPC would play a critical role in maintaining information about previous choices that will be needed for future credit assignment during interim decisions.

In the current study, we test these hypotheses using a learning task in which participants must track contingencies between specific choices and outcomes under conditions where choice-outcome transitions are direct following a delay, or indirect and involve an intervening decision. We show that in both conditions, the lOFC and HC reinstate representations of causal choices at the time of feedback. In the indirect condition, this information is critically dependent on representations of the causal choice maintained in a 'pending state' in lFPC, which predict subsequent reinstatement in lOFC and HC. Finally, we show that lOFC and HC code task-independent stimulus identity representations during feedback, suggesting a link between coding of a state's identity and precise credit assignment.

## Results

### Learning task with direct and indirect choice-outcome transitions

Participants completed a learning task in which they chose between two abstract shapes to obtain one of two distinct outcomes (gift cards to locally available stores rated to be approximately equally desirable). Each shape had a certain probability of leading to one gift card and the inverse probability of leading to the other. These probabilities drifted over time but could be tracked based on the recent choice-outcome observations made in each trial (see *Figure 1—figure supplement 1* for probability trajectories and Bayesian model fitting). Participants were informed of how many points each gift card would yield on each trial by colored numbers on the top of the screen, and that these points changed randomly from one trial to the next (*Figure 1A*). They were further told that at the end of the experiment one trial would be selected at random to count 'for real'. That is, they would receive the gift card obtained on that trial with a value proportional to the number of points won. Thus, participants were incentivized to maximize their potential winnings on every trial by accurately tracking the probability that each shape would lead to each outcome, but not the history of reward amounts.

The task had two conditions which proceeded in a blocked fashion. In the 'direct transition' condition, participants saw the outcome of a choice after a delay period (*Figure 1B*). In the 'indirect transition' condition, participants did not see the outcome of their choice until *after* another choice had been made, requiring them to delay assigning credit to the initial choice until the appropriate outcome was observed (*Figure 1C*). Participants were instructed about which condition they were in with a screen displaying 'Your latest choice' in the direct transition condition, and 'Your previous choice' in the indirect condition. Finally, at the beginning of each block participants viewed each of the two abstract shapes and two outcome stimuli in a random order, without making decisions or observing outcomes. This 'template' block allowed us to measure neural responses to stimuli independently of the learning task.

### Predicting current choice based on previous choice-outcome relationships

To test whether participants were using the structure of each condition to appropriately assign credit to causal choices, we performed a multiple logistic regression analysis testing the influence of previous choice-outcome combinations on the current choice. For each participant, independently in each condition, we constructed a GLM that predicted the current choice as a function of nine different combinations of previous choices and outcomes (*Equation 1*). For example, the first regressor predicted the current choice based on the previous choice and the previous outcome (trial *t-1*). These values were coded as 1 if the past choice led to the currently desired outcome, assumed to be the outcome with the largest monetary point value on the current trial, and −1 if it did not (results were virtually identical

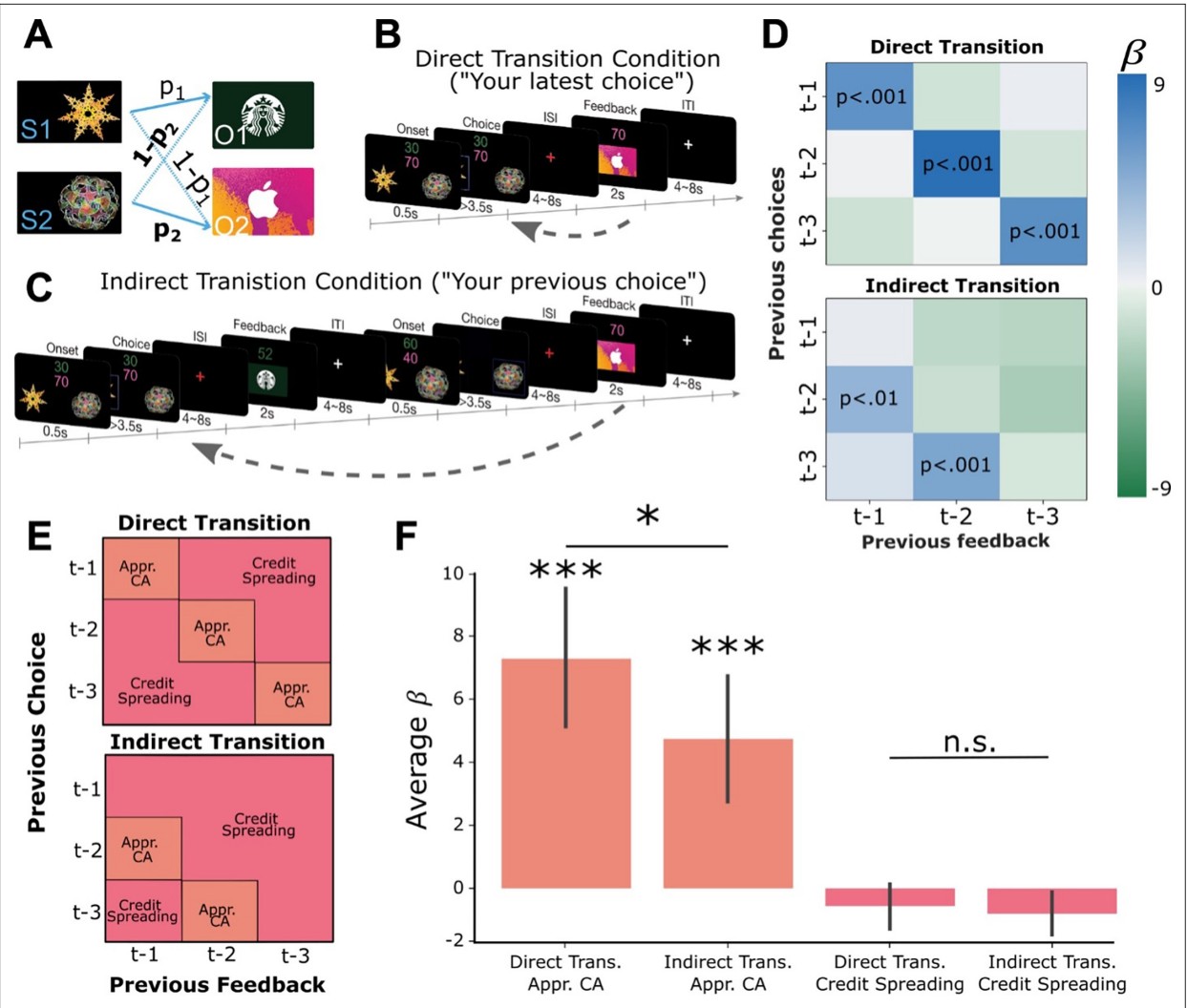

**Figure 1.** Learning Task Design and Behavioral Results. (**A**) Two abstract shapes were probabilistically related to each of two outcome identities by independent transition probabilities $p_1$ and $p_2$. (**B**) Schematic of the direct transition condition. Participants chose one of the two shapes on each trial based on two pieces of information: their estimates of the probability that each would lead to either outcome identity (gift cards) and the randomly generated number of points they could potentially win if that outcome was obtained. The color of each number indicated the identity of the outcome on which that number of points could be won. In the example, green indicates the number of points for the Starbucks gift card, while pink indicates the number of points for iTunes. Next, participants observed the outcome of their choice (the gift card and amount) after a delay. (**C**) Schematic of the indirect transition condition. Same as (**B**) except that after participants made their choice they transitioned into another independent decision. After this second decision was made, participants observed the outcome of their first decision. (**D**) Results of logistic regression analysis predicting the current choice based on previously observed choice-outcome relationships. Each cell represents the combination of a previously observed choice with an observed outcome. The color of each cell shows the value of beta estimates for each combination of previous choice and observed outcome, averaged across participants. Positive values indicate that the choice-outcome pair predicted choosing the same shape again when that shape previously led to the currently desired outcome. (**E**) Theoretical decomposition of the matrix in (**D**) into groups of cells which reflect 'appropriate credit assignment' given the task structure (orange) and 'credit spreading' (pink). (**F**) Mean (± SEM) of beta coefficients for specific choice-outcome combinations averaged across the groupings of cells shown in E for each condition. See *Figure 1—figure supplement 1* for model outputs and Bayesian model fitting.

The online version of this article includes the following figure supplement(s) for figure 1:

**Figure supplement 1.** Follow up behavioral analyses.

if we used the participant-specific indifference point (a) to define the desired outcome instead (see *Equation 9*)). The second regressor predicted the current choice based on the previous choice (*t-1*) and the outcome received two trials in the past (*t-2*), and so on for all nine combinations of previous choices and outcomes covering the previous three trials.

In the direct transition condition, we observed significant positive effects along the diagonal of the matrix ($choice_{t-1} * outcome_{t-1}$: β=6.09, $t(19)$ = 4.81, p<0.001; $choice_{t-2} * outcome_{t-2}$: β=8.78, $t(19)$ = 5.41, p<0.001; $choice_{t-3} * outcome_{t-3}$, β=6.76, $t(19)$ = 4.16, p<0.001; **Figure 1D**), indicating that participants assigned credit for each outcome to the choice made in same trial. In the indirect transition condition, current choices were significantly predicted by the most recently observed outcomes combined with choices made in the trial previous to those outcomes ($choice_{t-2} * outcome_{t-1}$: β=4.20, $t(19)$ = 2.92, p<0.01; $choice_{t-3} * outcome_{t-2}$: β=5.07, $t(19)$ = 4.75, p<0.001). Furthermore, the mean of the β-values which reflect appropriate credit assignment in each condition were significantly higher than the mean β-values which represented credit spreading (direct transition condition: $t(19)$ = 5.39, p<0.001, indirect transition condition: $t(19)$ = 4.34, p<0.001; **Figure 1E and F**). Follow-up analysis showed that participants' choices in each trial integrated expectations about the probability of receiving a particular outcome and its magnitude and did not rely on estimates of a cached option value (**Figure 1—figure supplement 1**). These results show that participants used the appropriate task-structure when assigning credit for observed outcomes in each condition.

Next, we compared the relative precision of credit assignment between our behavioral conditions, where we predicted credit assignment would be less precise in the indirect transition condition compared to direct transition condition, owing to additional task complexity. We found that β-values representing appropriate credit assignment in the direct transition condition were higher than those in the indirect transition condition ($t(19)$ = 1.81, p<0.05). However, β-values in cells that represent credit spreading in the direct transition condition were not significantly lower than those in the indirect transition condition ($t(19)$ = 1.11, p=0.14). These results indicate that credit assignment was less precise in the indirect transition condition compared to the direct transition condition, despite each being appropriate for the respective task structure overall.

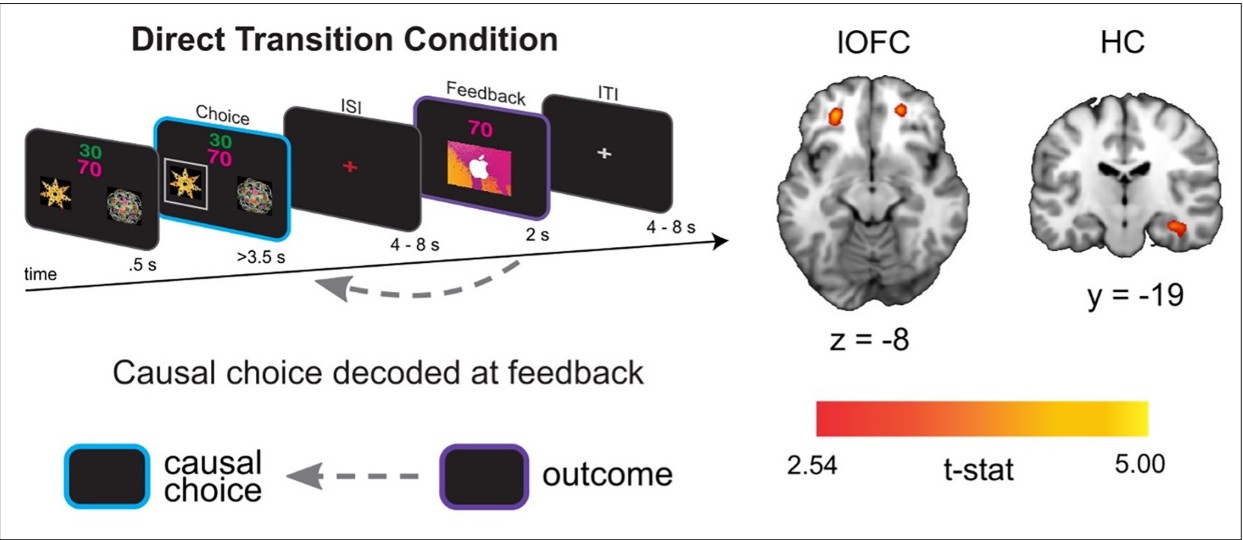

**Figure 2.** lOFC and HC carry representations of the causal choice when viewing outcomes. Left side shows the analysis scheme for decoding representations of the causal choice at feedback in the direct transition condition. An SVM decoder was used to differentiate trials at the time of the outcome (purple) based on the causal choice selected during the 'choice period' (cyan). The right side shows axial and coronal slices through a t-statistic map showing significant decoding in OFC and HC during feedback. For illustration, all maps are displayed at threshold of $t(19)$ = 2.54, p<0.01 uncorrected. All effects survive small volume correction in a priori defined anatomical ROIs. See **Figure 2—figure supplements 1–3** for ROI definition and **Figure 2—figure supplement 4** for power analysis.

The online version of this article includes the following figure supplement(s) for figure 2:

**Figure supplement 1.** Pre-selected anatomical ROIs.

**Figure supplement 2.** Functionally defined ROIs for in the direct transitions condition.

**Figure supplement 3.** Main effect of choice decoding accuracy at the time of feedback TFCE corrected in each run of the direct transition condition.

**Figure supplement 4.** Power analysis for Reinstatement Effect in the lOFC.

## Causal choice codes are reinstated in lOFC and HC when viewing the outcome of choices

For the direct feedback condition, our main hypothesis was that lOFC codes for the specific causal choice when participants view the outcome of their choice. We also reasoned that, due to the delay between choice and feedback, this lOFC choice code would be supported by choice reinstatement in the interconnected HC (*Barbas and Blatt, 1995*; *Wimmer and Shohamy, 2012*). We tested this hypothesis by training a linear support vector machine (SVM) to distinguish BOLD activity patterns at the time of feedback based on the previously chosen shape, cross-validated across scanning runs (see Methods for details on decoding procedure). We used a searchlight analysis within a priori defined ROIs (see *Figure 2—figure supplement 1*) for lOFC and HC to estimate decoding accuracy for each voxel within the ROI (*Kriegeskorte et al., 2008*).

We found evidence for choice decoding in the predicted network of regions. Specifically, we found significant and marginally significant decoding of the causal choice in left ([x,y,z] = [−26, 42,−8], $t(19)$ = 4.22, *pTFCE* <0.05 ROI-corrected using threshold-free cluster enhancement (TFCE) correction *Smith and Nichols, 2009*) and right ([x,y,z] = [24, 46, -8], $t(19)$ = 3.45, pTFCE = 0.081 ROI-corrected) lOFC, respectively (*Figure 2A*). A similar pattern was also apparent in the HC, where right HC showed significant decoding ([x,y,z] = [36, -20, -16], $t(19)$ = 4.02, *pTFCE* <0.05 ROI-corrected), while left HC showed a marginal effect ([x,y,z] = [-22,−10, −24], $t(19)$ = 2.86, pTFCE = 0.080 ROI-corrected). Together, these results show that the lOFC and HC represent the causal choice at the time when credit is assigned in the direct condition of our task.

## Pending item representations in lFPC during indirect transitions predict credit assignment in lOFC

The indirect transition condition allowed us to test whether similar reinstatement mechanisms, as described above, support credit assignment when choice-outcome transitions are punctuated by interim decisions. We anticipated that the structure of the indirect transition condition would render credit assignment more difficult compared to the direct transition condition; a prediction borne out by our behavioral analysis of learning (*Figure 1F*). Repeating the causal choice decoding analysis on this condition did not reveal a significant effect in any a priori defined ROI (all *pTFCE* >0.05 ROI corrected), nor did we find significant decoding elsewhere in the brain (all *pTFCE* >0.05 whole brain corrected). However, a key attribute of this condition is that causal choices must be held in a pending state during interim choices until a prospective outcome is observed. Thus, we reasoned that the fidelity of credit assignment at the time of feedback would be intimately related to the fidelity with which representations were maintained during the interim decision.

Following previous work suggesting that prospective representations of to-be-completed tasks are supported by lFPC (*Burgess et al., 2011*; *Koechlin and Hyafil, 2007*), we predicted that lFPC would hold causal choices in a 'pending state' when credit assignment needs to be deferred until the resulting outcome is observed. To test this hypothesis, we used a linear SVM to classify neural activity at the time of feedback based on the immediately preceding choice. Note that in this condition the immediately preceding choice is *not* the cause of the currently observed outcome, but is the cause of the outcome for which credit will be assigned in the next trial. We call this the 'pending causal choice'. Our analysis revealed a cluster of voxels specifically within the right lFPC ([x,y,z] = (28, 54, 8), $t(19)$ = 3.74, *pTFCE* <0.05 ROI-corrected; left hemisphere all *pTFCE* >0.1, *Figure 3A*), consistent with right lFPC coding for the pending causal choice at feedback time, precisely when the outcome of the prior choice causal choice needed to be evaluated.

To test whether pending choice information held in lFPC was directly related to the causal choice information coded during subsequent credit assignment we used an 'information connectivity' (IC) analysis, which seeks to identify how information is shared between brain regions (*Coutanche and Thompson-Schill, 2013*). Specifically, we tested the correlation between the fidelity of the previous choice representation when in a pending state, and the same causal choice representation during subsequent credit assignment. We began using an SVM to classify representations of the causal choice during the interim feedback period in voxels in the lFPC that were shown to code this information in our previous analysis (thresholded at $t(19)$ = 2.54, p<0.01).

Note that this relatively liberal threshold simply allows for the inclusion of more voxels for a statistically independent test in a left-out set of trials, thereby obviating selection bias. In a left-out set of

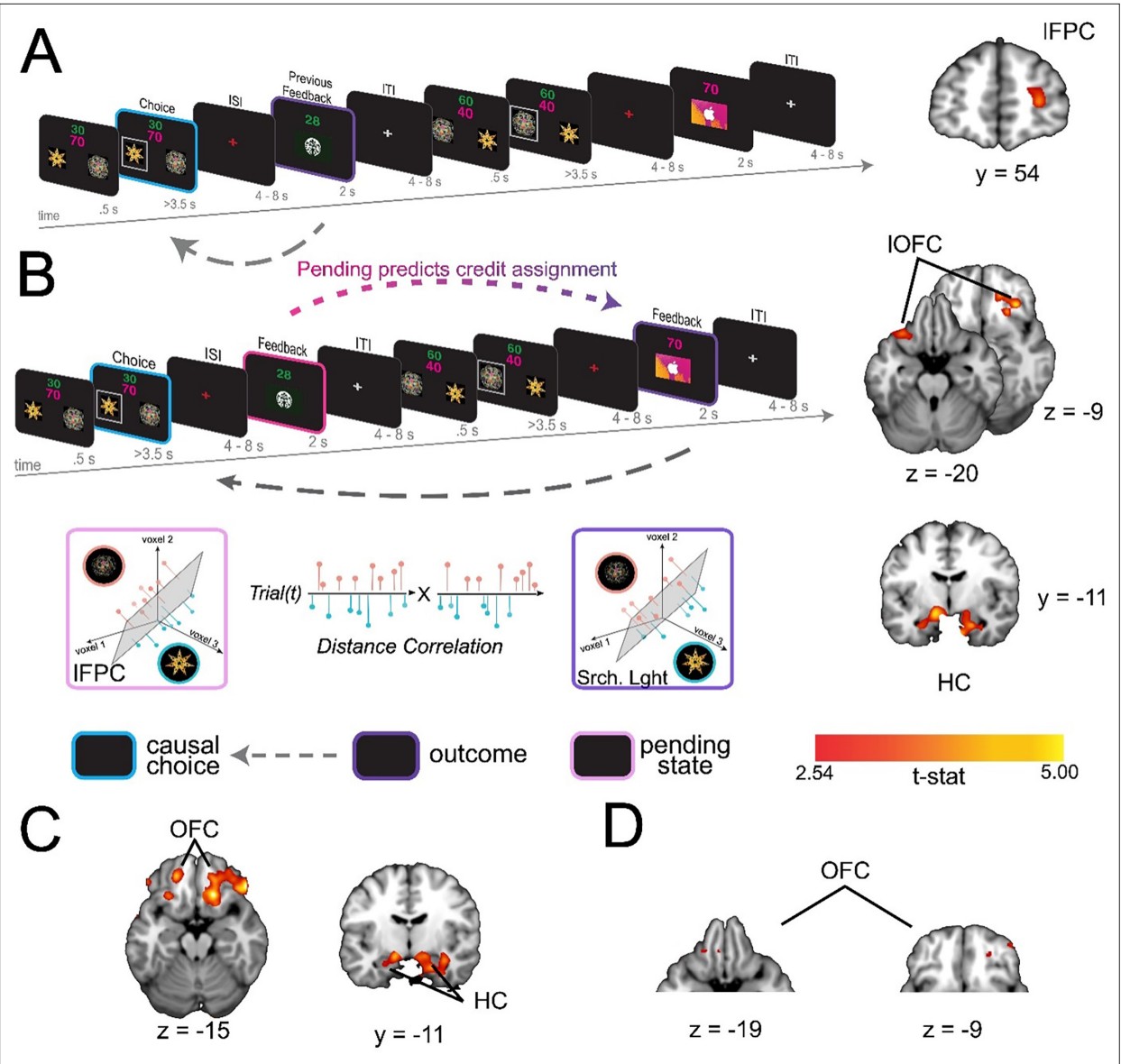

**Figure 3.** lFPC carries representations of the pending causal choice during indirect transitions and predicts credit assignment in the lOFC and HC.
(**A**) Left side shows the analysis scheme for decoding information about the causal choice in 'pending state' (pink) in the indirect transition condition.
We decoded information about the previous choice during the feedback period, during which the causal choice should be 'pending' credit assignment
in the next trial. The image on the right shows a coronal slice through a t-statistic map, showing significant decoding in lFPC. (**B**) The analysis scheme
for the information connectivity analysis which uses the trial-by-trial fidelity of causal choice representations in the 'pending state' (pink) to predict
the fidelity of these same choices when the outcome is observed (purple). The right side shows axial and coronal slices of a t-statistic map showing
effects in lOFC and HC. All maps are displayed using the same conventions as *Figure 2* and all effects survive small volume correction in a priori
defined anatomical ROIs (for whole brain analysis, see *Figure 3—figure supplement 2*). (**C**) Axial (left) and coronal (right) slices through a t-statistic
map showing the results of a control analysis in which we test the proportion of correct classifications of causal choice information in OFC and HPC at
the time of the outcome for trials in which the lFPC showed correct classification for the causal choice during pending trials. The proportion of correct
trials was compared to a permuted baseline of randomly drawn trials for each participant then combined over participants to create a t-statistic.
(**D**) Secondary control analysis in which we reran the classification analysis for causal choice information at the time of outcome, but only on trials where
lFPC was found to correctly decode pending causal choice information. Note that this test is different from A because we allowed the classifier to
create a new hyperplane separating categories for only those trials in which the lFPC decoding was 'correct'. For illustration, all maps are displayed
at a threshold of t(19)=2.54, p<0.01 uncorrected. All effects survive small volume correction in a priori and functionally defined anatomical ROIs. See
*Figure 3—figure supplements 1–2* for ROI definition and whole brain searchlight.

The online version of this article includes the following figure supplement(s) for figure 3:

**Figure supplement 1.** Significant information connectivity between lFPC and OFC in functionally defined ROI from direct transition condition.

**Figure supplement 2.** Exploratory information connectivity analysis for 'Indirect transition condition'.

trials, we calculated the distances between the estimated hyperplane and trial-level voxel activation patterns, and then signed these distances such that positive distances reflected 'correct' classifications and negative distances reflected 'incorrect' classifications. These signed distances allow us to relate both success in decoding information, as well as failures, between regions. Next, we applied the same method to quantify and sign the distances when decoding the same causal choices at the time of credit assignment – that is, when viewing the relevant outcome in the next trial. Finally, we correlated the decoding distances of causal choices in a pending state in lFPC with decoding distances of these choices during credit assignment in our lOFC and HC ROIs. This allowed us to assess whether the fidelity of pending causal choices representations in lFPC predicts the fidelity of representations during credit assignment in the lOFC and HC.

This analysis revealed strong IC between representations in lFPC at feedback on trial $t$ and the representations in lOFC and HC during feedback on trial $t+1$. Specifically, we found significant correlations in decoding distance between lFPC and bilateral lOFC ([x,y,z] = [-32,24,–22], $t$(19) = 3.81, [x,y,z] = [20, 38, -14], $t$(19) = 3.87, *pTFCE* <0.05 ROI corrected) and bilateral HC ([x,y,z] = [-28,–10, –24], $t$(19) = 3.41, [x,y,z] = [22, -10, -24], $t$(19) = 4.21, *pTFCE* <0.05 ROI corrected), *Figure 3C*. Subsequent analyses confirmed that this effect was due to these regions showing a significant increase in positive (correct) decoding in trials where pending information could be positively (correctly) decoded in lFPC, and not simply due to a reduction in incorrect information fidelity (see *Figure 3C and D*). This finding is consistent with the coding of the causal choice during feedback in lOFC and HC being dependent on that causal choice being faithfully maintained in a pending state in the lFPC.

## HC represents task-independent stimulus identity at feedback

Next, we tested whether the content of past choice coding at feedback includes a stimulus identity code that is reinstated during credit assignment. To test for task-independent representations of the causal stimuli, we trained a linear SVM to distinguish neural patterns evoked when participants passively viewed each shape in 'template trials' (see Methods). Importantly, these were presented outside the context of the learning task and were not connected to a specific action or outcome. We then tested the classifier on neural patterns evoked at the time of feedback during the learning task. This revealed significant decoding of the causal stimulus identity at the time of feedback when averaged across direct and indirect conditions, in the left HC (*Figure 4A*; [x,y,z] = [-26,–16, –16], $t$(19) = 5.20, *pTFCE* <0.001 ROI-corrected; right hemisphere all *pTFCE* >0.1). Follow-up analyses showed a marginally significant effect in the direct transition condition alone ([x,y,z] = [-24,–16, –14], $t$(19) = 3.41, pTFCE = 0.08 ROI-corrected), and a significant effect in the indirect transition condition alone ([x,y,z] = [-28,–16, –18], $t$(19) = 3.65 *pTFCE* <0.05). These results show that when observing an outcome, the HC reinstates task-independent representations of causal stimuli, suggesting a role for the HC in retrieving the causal stimulus identity during credit assignment.

We reasoned further that if the HC supports credit assignment by evoking task-independent identity representations, then the extent to which this information is coded in the HC should be intimately tied to behavioral estimates of credit assignment precision. Alternatively, identity representations in the HC might support credit assignment processes in lOFC, such that the extent to which this information is represented in lOFC is predictive of precise credit assignment. To test these predictions, we estimated each participant's overall credit assignment precision by correlating their pattern of β-values from the logistic regression models predicting choice with those of an 'ideal learner' (*Figure 4B*). The pattern for an ideal learner was taken to be 1 for any choice-outcome combination that reflected the true task structure, and 0 everywhere else. Higher correlations between these patterns meant that participants appropriately assigned credit to causal choices without attribution spreading to non-causal choices. We then correlated each participant's estimated credit assignment precision with the average decoding accuracy in HC and lOFC. We found that there was a significant correlation between credit assignment precision and decoding accuracy of the causal stimulus identity reinstatement in lOFC ([x,y,z] = [–24, 34,–16], $t$(19) = 3.24, *pTFCE* <0.05 ROI-corrected), but not HC (all *pTFCE* >0.09 ROI-corrected; *Figure 4C*). These results suggest that the extent to which identity information is reinstated in lOFC is directly related to the precision with which participants link appropriate choices and outcomes together.

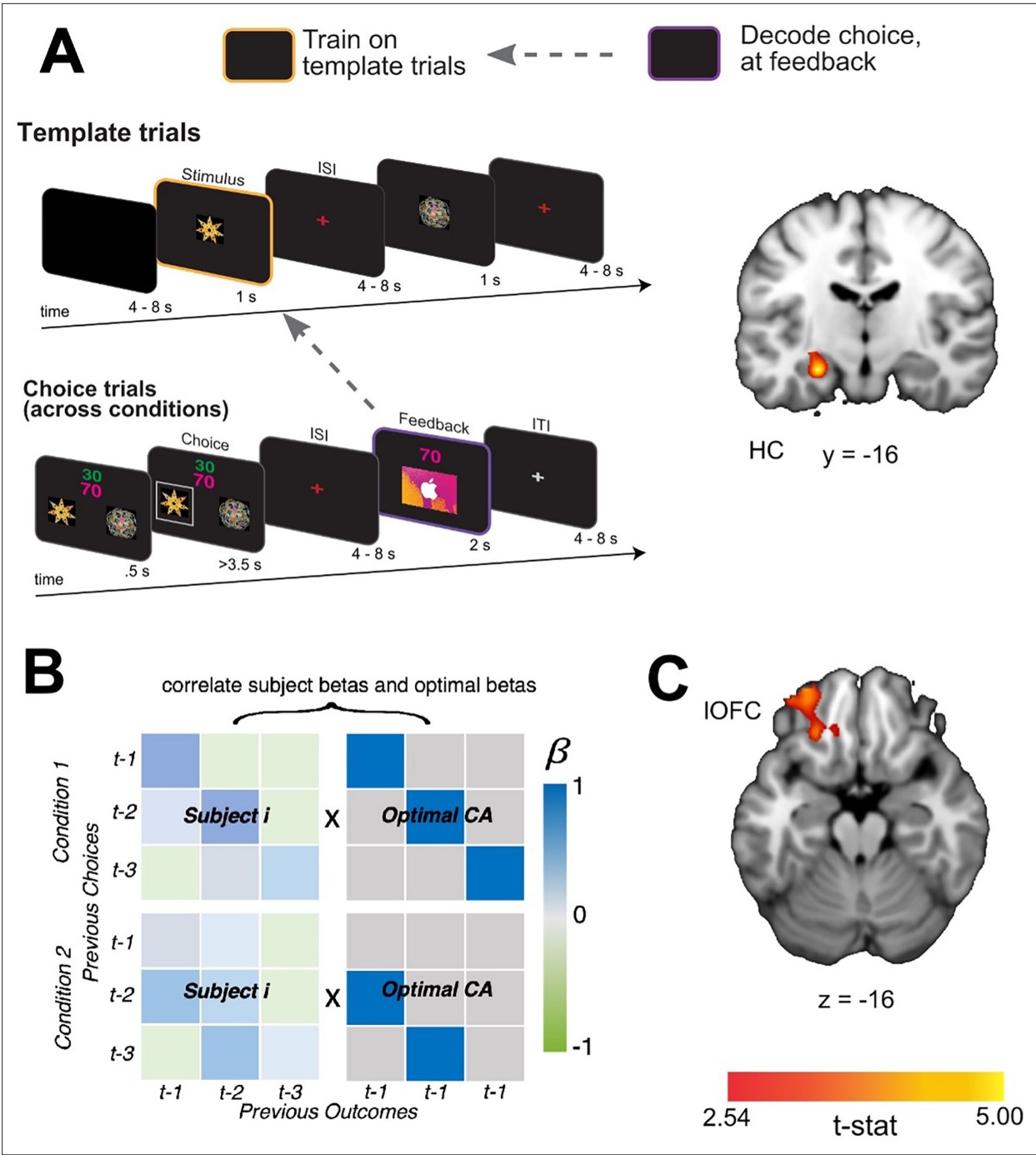

**Figure 4.** Task-independant representations of causal stimuli in HC at feedback. (**A - left**) Schematic of the decoding procedure. In task-independent 'template trials', participants passively viewed images corresponding to the two choice stimuli and two outcome stimuli in the main task (for more information see **Figure 4—figure supplement 1**). We used these trials to train an SVM to differentiate stimuli outside the task context and then tested for representations of the causal choice stimulus at the time of feedback during the learning task. (**A - right**) A coronal slice through a t-statistic map showing regions of the HC with significantly above chance decoding for the causal choice stimulus identity at the time of feedback, across conditions. In this figure, 'CA' refers to 'credit-assignment'. (**B**) Analysis scheme for generating each participant's overall credit assignment precision. β-values for each participant were taken from the behavioral model predicting current choices given all combinations of the previous three choices and outcomes (**Equation 1**). Each participant's pattern of β-values (left side matrices) were correlated with a matrix representing an optimal pattern of regression betas given the task structure (right side matrices). The optimal matrix was a binary matrix with ones where credit should be assigned for a given outcomes and zeros everywhere else. (**C**) Axial slice through a t-statistic map showing regions where decoding of the stimulus identity was significantly correlated with estimates of credit assignment precision. All maps are displayed using the same conventions as **Figure 2** and all effects survive small volume correction in a priori defined anatomical ROIs. See **Figure 4—figure supplements 1 and 2** for catch and bonus trials definition.

*Figure 4 continued on next page*

*Figure 4 continued*

The online version of this article includes the following figure supplement(s) for figure 4:

**Figure supplement 1.** Depiction of catch trials.

**Figure supplement 2.** Depiction of bonus trials.

## Discussion

Flexible decision making in dynamic environments requires an ability to learn choice-outcome relationships across prolonged delays, which may often be punctuated by interim decisions. Understanding how the brain assigns credit for specific outcomes, and forges connections with their causal choices, is essential for models of learning and decision-making that seek to explain how organisms implement such goal-directed behaviors. The current study reveals critical roles of the lOFC and HC in such credit assignment by showing that these regions specifically represent the *causal* choice at the time the outcome is observed. Importantly, we show that when credit assignment must be delayed due to an intervening choice, representations of the causal stimulus are maintained in a 'pending state' in lFPC. The fidelity of these representations determines the strength of causal choice representations in lOFC and HC when the outcome is subsequently observed. Finally, we show that the content of representations in HC includes the task-independent stimulus identities of the causal choice at the time of feedback, and the extent to which these are also represented in lOFC predicts precise credit assignment. Together, these results show that lOFC and HC adaptively use the task structure to associate identity-specific representations of causal choices to their resultant outcomes during learning and provide novel evidence for interactions between learning systems and lFPC in elaborated task structures which emulate real-world complexity.

Our finding that the lOFC instantiates a representation of the causal stimulus at the time of feedback contributes to a broader literature concerning the role of the lOFC in credit assignment. Previous research has shown that monkeys with lOFC lesions exhibit deficits in appropriately assigning credit to causal choices (*Walton et al., 2010*). Similarly, activity in human lOFC has been consistently associated with learning about contingencies between choices and rewards (*Boorman et al., 2016*; *Jocham et al., 2016*; *Lamba et al., 2023*; *Noonan et al., 2017*; *Witkowski et al., 2022*). We add to this literature by showing that the lOFC and HC contain specific multivariate patterns for inferred causal choices when an outcome is observed, suggesting that these regions are involved in updating links between choices and outcomes. Our results from the 'indirect transition' condition show that these patterns are not merely representations of the most recent choice but are representations of the *causal* choice given the current task structure, and may exist alongside representations of the task structure, in the lOFC and elsewhere (*Boorman et al., 2021*; *Park et al., 2020*; *Schuck et al., 2016*; *Seo and Lee, 2010*). These findings highlight a key role for the lOFC and HC in creating links between causal states and goal-states (*Boorman et al., 2021*; *Gardner and Schoenbaum, 2021*; *Howard and Kahnt, 2021*; *Wang and Kahnt, 2021*), and suggest that these regions use the specific task structure to construct causal associations between states.

While our study was designed to focus on the complexity of assigning credit in tasks with different known causal structures, another important component of real-world credit assignment is temporal ambiguity. To isolate the mechanisms which create associations between specific choices and specific outcomes, we instructed participants on the causal structure of each task, removing temporal ambiguity about the causal choice. However, our results are largely congruent with previously reported results in tasks that dissolved the typical experimental trial structure, producing temporal ambiguity, and which observed more pronounced spreading of effect, in addition to appropriate credit assignment (*Jocham et al., 2016*). Namely, this study found that activation in the lOFC increased only when participants received rewards contingent on a previous action, an effect that was more pronounced in subjects whose behavior reflected more accurate credit assignment. This suggests a shared lOFC mechanism for credit assignment in different types of complex environments. Whether these mechanisms extend to situations where the temporal causal structure is completely unknown remains an important question.

Importantly, we present novel evidence that representations of 'pending' causal choices are stored online in the lFPC and predict the strength of causal choice representations at the time of the outcome. Our results fit precisely with theoretical proposals of lFPC functions, which propose that this region

is involved in 'prospective memory' and tracking alternative behaviors or task sets during ongoing behaviors which may be returned to in the future (*Boorman et al., 2009*; *Burgess et al., 2011*; *Koechlin and Hyafil, 2007*; *Tsujimoto et al., 2011*). In the 'indirect transition' condition, participants needed to delay assigning credit when the first outcome was presented but return to this process when a prospective outcome was observed in the future. We show that when participants viewed outcomes for an unrelated choice, the lFPC held the content of the pending causal choice. These 'pending' representations predicted the strength of subsequent causal choice representations in lOFC and HC during the next feedback period, replicating the same network we observed in the 'direct transition' condition. The results extend prior work by showing that lFPC activity not only reflects statistics related to the evidence favoring pending options (*Badre et al., 2009*; *Boorman et al., 2009*; *Boorman et al., 2011*; *Donoso et al., 2014*), but the *content* of information held in a pending state. One interpretation of these results is that the lFPC actively protects information about causal choices when potentially interfering information must be processed. Future studies will be needed to determine if the lFPC's contributions are specific to these instances of potential interference, and whether this is a passive or active process. Nonetheless, the findings provide new evidence for the involvement of the lFPC in learning within complex task structures where the transitions between choices and outcomes are indirect - structures which abound in the real world.

Although we show evidence that lFPC is involved in maintaining specific content about causal choices during interim choices, the limited temporal resolution of fMRI makes it difficult to tell if other regions may be supporting the learning processes at timescales not detectable in the BOLD response. Thus, it is possible that the network of regions supporting credit assignment in complex tasks may be much larger. Our results provide a critical first stem in discerning the nature of interactions between cognitive subsystems that make different contributions to the learning process in these complex tasks.

A revealing aspect of our study was the inclusion of 'template' trials, which allowed us to measure task-independent neural responses to the stimuli used during the learning task. By training a classifier to decode stimulus representation during passive viewing, we were able to test which regions of the brain coded the specific stimulus identity of the causal choices during credit assignment. Consistent with previous accounts of hippocampal involvement in associative learning and inference (*Barron et al., 2020*; *Kurth-Nelson et al., 2015*; *Luettgau et al., 2020*; *Mack and Preston, 2016*; *Ranganath and Ritchey, 2012*; *Schuck and Niv, 2019*; *Wimmer and Shohamy, 2012*), we found significant decoding of task-independent choice identities in HC across participants in both direct and indirect conditions. This suggests that the HC retrieves a representation of the stimulus identity to bind together outcomes with causal choice information at the time of credit assignment, supporting the idea that the HC is involved in linking together previous experiences of sensory information (*McClelland et al., 1995*). Interestingly, recent work has shown the HC neuronal ensembles code a veridical representation of stimulus identities and predicted outcomes, which are critical to inference-guided choices (*Barron et al., 2020*). Together, these findings imply that a state's identity relationships constructed during credit assignment in the HC may be critical for future simulation of state-to-state transitions during outcome-guided inferences.

Interestingly, we found that the strength with which a stimulus identity can be decoded in the lOFC was correlated with behavioral measures of credit assignment, but not in HC. Recent work has shown that synchronized theta oscillations in macaques support information transfer from HC to the lOFC during value learning (*Knudsen and Wallis, 2020*). Disrupting these signals leads to learning deficits, suggesting that these regions work in concert to support value learning based on a relational cognitive map of the task. This synchrony between regions also finds support in human work showing strong functional connectivity and shared information between the anterior medial temporal cortex and OFC (*Barnett et al., 2021*; *Bouffard et al., 2021*; *Ranganath and Ritchey, 2012*). In our task, it is possible that while the HC coded task-independent identities of causal stimuli, the extent to which this information was transferred to, and represented, in the lOFC determined the efficacy of credit assignment. Future studies using methods with higher temporal resolution can elaborate on this idea by testing whether the HC and lOFC also share coherent stimulus identity information that is likewise channeled via theta phase coupling at the time of outcome, and how this information influences the credit assignment process.

In conclusion, we find that the lOFC and HC are critical to using model-based knowledge for efficiently forging links between outcomes and causal choices. Further, we show that in complex

tasks where choice-outcome transitions may be interrupted, this credit assignment network relies on interactions with the lFPC, which maintains 'pending' representations of causal stimuli during the interim decision. Collectively, these findings make a novel contribution to our understanding of credit assignment in the brain by illuminating the neural mechanisms which underlie linking causal choices to outcomes in complex, real-world tasks.

# Methods

**Key resources table**

| Reagent type (species) or resource | Designation | Source or reference | Identifiers | Additional information |
|---|---|---|---|---|
| Software | MATLAB | MathWorks | https://www.mathworks.com; RRID:SCR_001622 | Matlab2018a |
| Software | Presentation | Neurobehavioral Systems | http://neurobs.com; RRID:SCR_002521 | Version 18.1 |
| Software | LIBSVM | *Chang and Lin, 2011* | http://www.csie.ntu.edu.tw/~cjlin/libsvm; RRID:SCR_010243 | |
| Software | MarsBaR | *Brett et al., 2002* | http://marsbar.sourceforge.net/; RRID:SCR_009605 | Ver. 0.44 |

## Participants

Twenty participants (11 females; 9 males; mean age = 23.5) were recruited from the general population around University College London to participate in the study. This sample size was commensurate with previous studies similar in design (*Boorman et al., 2016*; *Howard et al., 2015*; *Jocham et al., 2016*). Using an independent, unpublished data set, we conducted a power analysis for the desire neural effect in lOFC. We found that this number of participants had 84% power to detect this effect (see *Figure 2—figure supplement 4*). Participants were paid £10 and obtained a gift card of various amounts depending on their performance in the task. None of the participants reported a history of neurological or psychiatric disorder. All participants spoke fluent English and had normal or corrected-to-normal vision. The study was approved by the UCL Research Ethics Committee (Project ID Number: 3450/002), and all participants gave written informed consent.

## Task design

### Learning task

Participants completed a learning task in which they tracked associations between abstract shapes and specific reward identities (gift cards to two different stores), which were rated for approximately equal desirability. In each trial, participants selected one of two abstract shapes, which were randomly presented on either the left or right side of the screen. Decisions were based on two pieces of information: (1) inferred estimates of the probability that a particular shape would lead to each gift card based on the history of previous trials, and (2) the point value of each gift card on the current trial (*Figure 1A–C*). Participants were informed prior to starting the task that one of the trials would be chosen at random to count 'for real' at the end of the experiment. For this trial, they would receive money on the awarded gift card that was commensurate with the number of associated points (number of points divided by four). Point values for each outcome were presented as two numbers at the top of the screen, with the color of each number indicating the associated gift card identity. Their position relative to each other (top or bottom) was determined randomly on each trial.

Each shape had a specific probability of leading to each outcome and an inverse probability of leading to the other outcome. For example, shape 1 (S1) might lead to a Starbucks gift card with probability $p_1$ and to an iTunes gift card with probability $1 - p_1$. Shape 2 (S2) would lead to the same outcomes but with independent probabilities $p_2$ and $1 - p_2$, respectively. These true probabilities would drift independently over the course of the experiment, meaning that information about outcome probabilities could not be shared across shapes. On any given trial, the number of points that could be won for each gift card ranged from 20 to 100, with a minimum difference of at least 15 points. Although these magnitudes were predetermined, participants were told they were randomly generated at the

beginning of each trial and that it was not useful to track them (Pearson correlation between magnitudes in trial n and n+1 was less than.2). Instead, to maximize rewards, participants had to track the probability that a shape led to each outcome and combine this with the reward magnitudes associated with each outcome on the current trial.

Each trial began with viewing the two possible choices for 0.5 s, during which selection was not possible. They then had 3.5 s to make their selection between the two options. The selected shape was highlighted for 0.5 s, before proceeding to the interstimulus interval (ISI), which lasted for a randomly selected duration between 4 s and 8 s. The outcome was then presented for 2000ms before a jittered inter-trial-interval (ITI) of 4s to 8s.

Participants did not have any prior knowledge about choice-outcome associations or how quickly these associations might change, but they knew that they could change throughout the task. Therefore, participants needed to infer both the current associative contingency for each shape and when these contingencies changed from their history of choices and observed outcomes.

## Template task

Each run of the scanning session began with a 'template task'. In this task, participants passively viewed a sequence of all four stimuli (two shapes and two gift cards), individually presented in random order. To ensure that participants were paying attention during passive viewing, they were presented with four 'catch trials' which occurred at random between images (see *Figure 4—figure supplement 1*). In catch trials, all four stimuli were presented simultaneously, and participants were asked to indicate which stimulus had just been presented (see *Figure 4—figure supplement 2*). Participants were told they could earn an additional £10 on the selected gift card if they responded correctly. However, they would be deducted £1 for each incorrect response or for not making responses in time (max response time = 3 s). Average accuracy for these catch trials was generally high (mean = 0.75, std = 0.15). Participants viewed each item for 1 s followed by a 2.5 s ISI.

## Stimuli

Two visually distinct abstract shapes were used as choice objects. These shapes were randomly assigned to serve as S1 or S2 for each participant. The two gift cards were chosen to serve as reward identities during the experiment from six different possible gift cards (iTunes, Argos, Blackwells, Marks & Spencers, Boots, and Starbucks). Each participant rated the six gift cards on a scale from 0 (not preferable) to 100 (extremely preferable). The two gift cards were selected to have the minimal difference in ratings among the highest rated gift cards. This was done to prevent a strong preference for one outcome over the other. All stimuli were presented on a computer running Presentation software (Version 18.1, https://www.neurobs.com/).

## Task-schedule and procedure

We generated a reward schedule that predetermined the outcome obtained for each choice on each trial, but this schedule was unknown to the participants. We optimized the schedule such that an ideal Bayesian learner (see Bayesian Computational model) would choose each shape and receive each outcome approximately an equal number of times (percent of overall trials where S1 was chosen was between 42% and 57%). This was done to reduce the potential for sampling bias in planned multivariate analyses. The schedule of outcomes for each shape was generated with independently drifting probabilities so participants could not learn anything about one shape from observing the outcome of the other shape (see *Figure 1—figure supplement 1*).

Participants completed three scanning runs in one session. The first two runs began with the template task, which was followed by the learning task (37 trials of the direct transition condition, then 37 trials of the indirect transition condition). The third run consisted of only the template task. The learning task began with instructions stating, 'Your latest choice', indicating that participants were in the direct transition condition. After 37 trials, a second instruction screen showed 'Your previous choice' indicating that participants were about the start indirect transition condition. Participants knew that in the indirect transition condition, the first outcome observed was not linked to any choice.

In each run, we included three 'bonus trials' (two in the direct transition condition and one in the indirect transition condition), distributed throughout choice trials, which occurred between a choice and the outcome. Participants were shown the two gift cards on either side of a question mark

and were given the chance to predict which outcome they would receive in the upcoming feedback period. For each correct gift card prediction, they received an additional £3 on the gift card they would receive at the end.

## Behavioral training

Prior to each scanning session, participants completed a shortened (76 trials) behavioral training session. In the training session, participants completed a practice version of the choice task, which had a unique reward schedule. Prior to the practice trials, participants were verbally given a 'comprehension quiz' to verify they understood key elements of the task, such as the difference between choice-outcome transitions in each condition. Finally, the distribution of ISI and ITI durations for this session was constrained to 2s to 4s.

## MRI data acquisition and preprocessing

The brain images were acquired using a 32-channel head coil from a 3 Tesla Siemens Trio scanner. We used a T2*-weighted echo-planar imaging (EPI) sequence to collect 43 2 mm slices in ascending order, with 1 mm gaps. The in-plane resolution was of 3x3 mm, with a repetition time (TR) of 3.01 s and echo-time (TE) of 70ms. We set the slice angle to a 30 degree tilt relative to the rostro-caudal axis to minimize signal loss from the lOFC (*Weiskopf et al., 2006*) and applied a local z-shim with a moment of –0.4 mT/m to the OFC. The first five volumes of each block were discarded to allow for T1 equilibration effects. For accurate registration of the EPI to a standard space, we acquired a T1-weighted anatomical scan with a magnetization-prepared rapid gradient echo sequence (MPRAGE) with a 1×1 × 1 mm resolution. Finally, to measure and correct for geometric distortions due to susceptibility-induced field inhomogeneities, a whole-brain field map with dual echo-time images (TE1=10ms, TE2=14.76ms, resolution 3×3 × 3 mm) was also acquired.

We performed slice time correction, corrected for signal bias, and realigned functional scans to the first volume in the sequence using a six-parameter rigid body transformation to correct for motion. Images were then spatially normalized by warping participant-specific images to the reference brain in the MNI (Montreal Neurological Institute) reference brain and smoothed using an 8 mm full-width at half maximum Gaussian kernel. Pre-processing was done in SPM12 (Wellcome Trust Centre for Neuroimaging, http://www.fil.ion.ucl.ac.uk/spm) using Matlab 2018a.

## Quantification and statistical analyses

### Regression analysis

To test whether participants showed a behavioral effect of learning on choice, we fit logistic regression models estimating the influence of past choice-outcome observations on choices in the current trial $t$. The regression model included the effect of the past three choices ($C_{t-n}$) in combination with the past three observed outcomes ($O_{t-n}$). For example, $C_{t-1}O_{t-1}$ represents the influence of the most recent choice and the most recent outcome on the current choice. The model estimates the probability of making choice C on trial $t$ given all nine combinations of previous choices and outcomes:

$$p\left(\text{choice} = C\right)_t = \beta_0 + \beta_1 C_{t-1}O_{t-1} + \beta_2 C_{t-2}O_{t-1} + \beta_3 C_{t-3}O_{t-1} + \beta_4 C_{t-1}O_{t-2} + \beta_5 C_{t-2}O_{t-2} +$$
$$\beta_6 C_{t-3}O_{t-2} + \beta_7 C_{t-1}O_{t-3} + \beta_8 C_{t-2}O_{t-3} + \beta_9 C_{t-3}O_{t-3} + \epsilon \tag{1}$$

The value of $C_{t-n}$ was taken to be 1 if they chose shape S1 on trial $t-n$ and –1 if they chose S2. The value of $O_{t-n}$ was taken to be 1 if the outcome on trial *t-n matched the currently desired outcome*, on trial $t$, and –1 if it did not. The currently desired outcome was assumed to be the outcome with the largest point value in each trial. Thus, the value of $C_{t-n}O_{t-n}$ for each trial was 1 if choice C led to the currently desired outcome $n$-trials back and –1 if it did not:

$$C_{t-n}O_{t-n} = \begin{cases} 1 & ifC_{t-n}led\ to\ the\ currently\ desired\ outcome \\ -1 & ifC_{t-n}led\ to\ the\ currently\ undesired\ outcome \end{cases} \tag{2}$$

We fit separate regression models for each condition in each run for every participant. We then averaged the resulting regression coefficients (β) across runs, resulting the participant specific influence of previous decisions on the current choice.

## Bayesian computational model

We used a Bayesian computational model to predict choices in each trial $t$ based on each participant's previously observed shape-outcome relationships (i.e. the estimated associative probability), and reward magnitudes in the current trial. We briefly describe the model here, but a full description can be found in **Behrens et al., 2007**; see also **Arulampalam et al., 2002** for a related model.

Since the true probability of the associative contingencies cannot be observed, the model estimated, in a Markovian fashion, the subjective belief that making a given shape ($S$) would lead to outcome 1 (O1), and to outcome 2 (O2) with the inverse probability:

$$p\left(S \rightarrow O1\right) = p_S$$
$$p\left(S \rightarrow O2\right) = 1 - p_S \tag{3}$$

where $p_s$ denotes the associative probability of a given shape $S$ leading to O1. On each trial ($t$) the model estimated the current value of $p_{st}$, based on the previous observations of outcomes $y_{1:\,t}$. We modeled beliefs about the likelihood of each contingency as a beta distribution over possible values of $p_{st}$:

$$\beta(p_{St}|V) \tag{4}$$

where $p_{st}$ is the mean of the beta distribution and $V = exp\left(v\right)$ describes the variance. A large value of $v$ means that the value of $p_{st}$ is likely to change in the next trial whereas low values of $v$ mean that it is unlikely to change. Here, $v$ is referred to as the 'volatility' because it controls the learning rate for shape-outcome associations. The change in the estimated volatility from previous trial to the current trial is controlled by $k$. This describes the model's belief that some level of change in the volatility is going to occur in the next trial. Because there are no constraints on values for $v_t$, this distribution can be modeled as a Gaussian:

$$p\left(v_t \vee v_{t-1}, K\right) = N\left(v_{t-1}, k\right) \tag{5}$$

After observing each piece of evidence about the contingency between shape S and the outcome, the estimate of each parameter could then be updated following Bayes rule

$$p\left(p_{St}, v_t, k\right) = p\left(y_t|p_{St}\right) \int \int \left[p\left(p_{St-1}, v_{t-1}, k|y_{1:\,t-1}\right) p\left(v_t|v_{t-1}, k\right) dv_{t-1}\right] p\left(p_{St}|p_{St-1}, v_t\right) dp_{St-1} \tag{6}$$

This gives us the three-dimension joint probability of the parameters. On each trial, the learner only needs to know the estimated contingency between a shape and outcome which is performed first by marginalizing over $v$ and $k$:

$$p\left(p_{St}\right) = \int \int p\left(p_{St}, v_t, k\right) dv_t dk \tag{7}$$

And then taking the mean of the resulting distribution.

$$\widehat{p_{St}} = \int p_{St} p\left(p_{St}\right) dp_{St} \tag{8}$$

For each participant, we initialized the model with a uniform prior over the entire parameter space. All integral computations are performed using numerical grid integration. We then used the prior belief in the associative contingencies $p_{St}$ to compute the expected value of each shape on each trial according to the following formula:

$$Ev_{St} = [\widehat{p_{St}} m_{O1t} \alpha] + [[1 - \widehat{p_{St}}] m_{O2t} [1/\alpha]] \tag{9}$$

where $\alpha$ was a free parameter and reflected a participant's preference for O1 over O2 (0< $\alpha$ <2), and $m_{O1t}$ and $m_{O2t}$ indicated the reward magnitudes of the outcome available in the current trial, $t$. We then measured the likelihood of each participants choice on each trial according to a SoftMax function:

$$p\left(choice = S1\right) = e^{bEv_{S1t}} \left(e^{bEv_{S1t}} + e^{bEv_{S2t}}\right)^{-1} \tag{10}$$

where the free parameter $b$, captured the level of sensitivity of choices to expected values (inverse temperature; $0<b<1$). Free parameters were fitted using Markov Chain Monte Carlo (see below).

## Value-based RL- model

This model estimated the value of each shape given the history of rewards received from choosing the shape. The value of each shape was initiated at 0, then updated using the following equation:

$$V\left(S_{xt}\right) = V\left(S_{xt-1}\right) + \delta\left(\alpha R_t - V\left(S_{xt-1}\right)\right) \tag{11}$$

where $R_t$ is the magnitude of the reward on trial $t$ and α is an individual difference term estimating a participant preference for one outcome over the other ($0< \alpha <2$). The learning rate ($\delta$) was estimated for each participant to capture the magnitude of the update ($0< \delta <1$). We entered these values into a SoftMax function to generate choice probabilities:

$$p\left(choice = S1\right) = e^{bV(S_{1t})}\left(e^{bV(S_{1t})} + e^{bV(S_{2t})}\right)^{-1} \tag{12}$$

where the free parameter $b$, captured the level of sensitivity of choices to expected values (inverse temperature; $0<b<1$). Free parameters were fitted using Markov Chain Monte Carlo (see below). Note that learning failures are not trivial to identify in our paradigm and model, because every choice is based on a participant's preference between gift card outcomes, and the ability of the computational model to accurately estimate participants' beliefs in the stimulus-outcome transition probabilities.

## Parameter estimates

The Bayesian learning model has two free parameters, α and $b$. The value RL-model had an additional parameter $\delta$. We fit these parameters independently for each participant using custom Markov Chain Monte Carlo (MCMC) code in MATLAB R2018a. Model parameters were bounded by the following: $[0<\alpha<2]$, $[0<b<1]$, $[0< \delta <1]$ and were initialized at $\alpha=1$ and $b=0.5$, $\delta=0.5$. Each model was fit to maximize the likelihood of a participant's choices given model estimates of the expected value of each choice on each trial (*Equation 10*; *Equation 12*).

## Multivariate decoding of causal choice and pending causal choice representations

Using multivariate pattern analysis (MVPA), we aimed to identify regions of the brain that coded knowledge of causal choices during the feedback period. To test this, we estimated the BOLD activity patterns during the feedback phase for each trial using unsmoothed preprocessed images. The feedback periods were modeled as boxcars that had a constant duration lasting 2000ms from the onset of the outcome presentation in each trial. The GLM also included regressors for the decision period (modeled as boxcars with a duration equal to RT) and template presentations (modeled as boxcars with a 1000ms duration). No parametric modulators were added. Each trial was labeled according to which shape was chosen during the choice period (either S1 or S2). For our analysis of 'pending' representations in the indirect transition condition, we linked these labels to the immediately following, interim feedback phase - a time when participants should be delaying credit assignment in anticipation of assigning credit in the next trial.

We used a searchlight procedure to identify regions of the brain that contained representations of the causal choice. Each searchlight consisted of a 5 x 5 × 5 voxel cube placed around a centroid voxel in the brain. Each centroid was required to have values in at least 10 of the surrounding voxels to be considered for further processing. The activity in each trial was standardized by z-scoring the β-values across voxels within each searchlight. The data were then split by blocks into training and test sets by run. We used LIBSVM (*Chang and Lin, 2011*) to fit linear classifiers with training data, which were subsequently used to classify data points from the test set. We iterated through this process for each of the two runs then computed the mean decoding accuracy (average proportion of correct classifications) across both classifiers. The mean decoding accuracy for each voxel was compared to a voxel-specific null distribution which was estimated by repeating this procedure while randomly assigning the labels for 100 permutations at each searchlight. The mean classification accuracy of this null distribution was subtracted off the classification accuracy of each searchlight to give us a measure

of how reliably information about the causal choices could be decoded above chance. The resulting maps were then spatially smoothed using a Gaussian kernel with full width at half maximum of 8 mm.

Group-level analyses were performed using a one-sample t-test on accuracy maps across participants (see *Group-level statistical inference*). We corrected for multiple comparisons over a priori defined ROIs in lOFC, HPC, and lFPC, and used functionally defined ROIs for lOFC in a data driven ROI analysis (see *Figure 2—figure supplements 1–3*).

To ensure that participants where we included valuable *catch trials* in the passive observing 'template task'. Participants were asked to report which image out of the four (2 gift cards and 2 stimuli) was the last one presented on the screen. They were endowed an extra £10 from which we removed £1 for every incorrect response. There were four catch trials per template run. –5. We corrected for multiple comparisons using small volume correction TFCE. The threshold for significance remained the same in all analyses (pTFCE <0.05).

## Multivariate analyses of information connectivity between regions

To test whether decoding of the causal choice at feedback in the indirect transition condition depended on the strength of 'pending' representations held during the interim trial, we tested whether the fidelity of representations of the pending causal choice in lFPC was associated with the fidelity of those same choices at the time of credit assignment (i.e. in the feedback phase of the next trial). We used the same decoding procedure mentioned above to classify voxel patterns at feedback in each trial, but additionally calculated the distance of each pattern from the hyperplane that divides categories. Distances were obtained using the equation specified on the LIBSVM webpage (https://www.csie.ntu.edu.tw/~cjlin/libsvm/faq.html). Patterns that are more distant from the hyperplane can be thought of as having higher fidelity, and those that are closer to the hyperplane as having less (*Schuck and Niv, 2019*). We then signed the distance of each point according to whether the predicted category label was correct (+for correct, – for incorrect).

First, we calculated trial-by-trial distance from the hyperplane when causal choice information was believed to be held in a 'pending' state, focusing on lFPC as our 'seed-region'. For this, we calculated the average distances for voxels within the lFPC that showed significant decoding of the pending choice during the interim feedback period ($t(19)=2.54$, p<0.01 uncorrected). This gave us a measure of the information about the pending item on each trial. We calculated the decoding strength of these same choices when the true outcome was shown, as a measure of the information about the causal choice during credit assignment. Here, we calculated distances for every 5x5 × 5 voxel cube using the same searchlight procedure we described above. Note that the decoding fidelity metric at each time point represents the decodability of *the same choice at different phases of the task.* These phases were separated by at least 10 s and 15 s on average, which can be sufficient for disentangling unique activity (*Mumford et al., 2012*; *Mumford et al., 2014*). We then correlated the decoding distance for representations in lFPC during 'pending' state and the decoding distance of those same choices at credit assignment. Thus, the correlation value between them gives us a measure of whether strong representations of pending causal choices in lFPC predict stronger representations at credit assignment.

To confirm that this correlation did not simply arise because the classifier in each region is 'less wrong' when the decoder in lFPC makes correct classifications (*i.e., all classifications were wrong, but the test region was less wrong)*, we performed two control analyses. First, we calculated the frequency of correct classifications for the subset of trials in which lFPC also showed correct classifications. We then compared the frequency of correct classifications to a permuted baseline frequency by randomizing trial distances in the searchlight then recomputed the frequency of correct classifications. We subtracted the mean of the randomized baseline from the true frequency of correct classifications. This gave us a measure of decoding accuracy in each searchlight when lFPC showed correct decoding accuracy. Our second control analysis involved rerunning the classification procedure (see *Multivariate analyses of credit-assignment and pending representations),* but only for trials in which the lFPC had already shown correct decoding of the causal choice in a pending state. Again, we compared the accuracy of the classifier in each searchlight to a randomized baseline frequency by randomizing trial labels and recomputing the accuracy of the classifier. The mean of the randomized distribution was then subtracted from the classification accuracy using the true labels.

Group-level analyses were performed by Fisher-z transforming the correlation values then using a one-sample t-test on each voxel. We corrected for multiple comparisons using TFCE correction on the resulting volumes within a priori *defined* ROIs. The same thresholds were applied for group level statistical correction (pTFCE <0.05).

## Multivariate analyses of identity codes during credit assignment

To test whether the task-independent identity of the causal choice was reinstated during feedback, we trained a linear SVM to decode representations of causal choice stimuli but trained the classifier during periods when participants passively viewed the stimuli outside of the task context (see 'Template trials'). In each condition the SVM was trained on all the trials of the three template runs and tested during the feedback period of the learning task. For each participant and in each trial, we estimated the BOLD activity patterns using the same GLM as described above (see 'Multivariate decoding of causal choice and pending causal choice representations'). Further, we used the same procedure in which we randomly permuted the training labels 100 times to create a null distribution of decoding accuracy. We then averaged decoding accuracy over runs and subtracted the mean of the null distribution from the true decoding accuracy of the classifier.

To test for associations between credit assignment precision and causal choice identity decoding accuracy, we first generated estimates of credit assignment precision based on each participant's behavior during the task. For each participant we created a behavioral matrix, which included β-values from nine combinations of possible choice-outcome relationships used to assign credit when an outcome is observed (see 'Regression model'). For the direct transition condition, values along the diagonal of this matrix represent appropriate credit assignment given the task structure and should have high positive values if the participant is assigning credit precisely. All other values should be near 0. A similar matrix can be generated for the indirect transition condition, but appropriate for the causal structure of this condition (see *Figure 1E*). Next, we created a comparison matrix based on an idealized learner, with values of 1 in each cell that represented appropriate credit assignment for the condition, and values of 0 for non-causal relationships. We then correlated each participant specific behavioral matrix with the comparison matrix. High correlation values represent more precise credit assignment, and the average across conditions was taken to be a measure of the overall credit precision in the learning task. We then regressed each participant's overall credit precision estimate against voxel-level decoding accuracy across participants. We corrected for multiple comparisons using TFCE correction to volumes within pre-defined ROIs. The same thresholds were applied for group-level statistical correction (pTFCE <0.05).

## Group-level statistical inference

Group-level testing was done using a one-sample t-test (df = 19) on the cumulative functional maps generated by the first-level analysis. All first-level maps were smoothed prior to being combined and tested at the group level. To correct for multiple comparisons, we first extracted voxels from each ROI in each participant's first-level activation map, then applied Threshold-Free Cluster Enhancement (TFCE) which uses permutation testing and accounts for both the height and extent of the cluster (*Smith and Nichols, 2009*). All parameters were set to default parameters (H=2, E=0.5) and used 5000 permutations for the analysis. We report effects that surpassed a pTFCE <0.05 threshold in each ROI.

## Region of interest selection

Regions of interest in the prefrontal cortex were generated from anatomically defined regions with unique functional connectivity fingerprints (*Neubert et al., 2015*). The lOFC ROIs corresponded to bilateral area BA11 (indexes 9 and 30). We included these regions because they have been previously implicated in credit assignment for causal choices, particularly in similar contingency learning tasks (*Boorman et al., 2016*; *Jocham et al., 2016*). For the lFPC, we used indexes 14 and 35. All of these ROIs were threshold at 60% inclusion criteria, although our results did not qualitatively change at different thresholds. Finally, we used a priori anatomically defined bilateral HC ROIs to test for effects in hippocampus (*Yushkevich et al., 2015*). These ROIs are illustrated in *Figure 2—figure supplement 1*.

## Acknowledgements

Funding was provided by a Sir Henry Wellcome Postdoctoral Fellowship and NSF CAREER Award (1846578) to EDB, a Senior Research Fellowship from the Wellcome Trust and an award from the James S McDonnell Foundation to TEB, and a Principal Research Fellowship from the Wellcome Trust to RJD. This work was also in part supported by the Intramural Research Program at the National Institute on Drug Abuse (ZIA DA000642). The opinions expressed in this work are the authors' own and do not reflect the view of the NIH/DHHS.

## Additional information

### Competing interests

Timothy EJ Behrens: Editor-in-Chief, *eLife*. The other authors declare that no competing interests exist.

### Funding

| Funder | Grant reference number | Author |
| --- | --- | --- |
| National Science Foundation | 1846578 | Erie Boorman |
| Intramural Research Program at the National Institute | ZIA DA000642 | Phillip P Witkowski |
| Wellcome Trust | Principal Research Fellowship | Raymond J Dolan |
| Wellcome Trust | Senior Research Fellowship | Timothy EJ Behrens |
| James S McDonnell Foundation | | Timothy EJ Behrens |
| Sir Henry Wellcome Postdoctoral Fellowship | | Erie Boorman |

The funders had no role in study design, data collection and interpretation, or the decision to submit the work for publication. For the purpose of Open Access, the authors have applied a CC BY public copyright license to any Author Accepted Manuscript version arising from this submission.

### Author contributions

Phillip P Witkowski, Formal analysis, Investigation, Visualization, Writing – original draft; Lindsay JH Rondot, Data curation, Formal analysis, Investigation, Writing – original draft; Zeb Kurth-Nelson, Formal analysis, Investigation; Mona M Garvert, Investigation, Writing – review and editing; Raymond J Dolan, Supervision, Funding acquisition, Investigation, Project administration, Writing – review and editing; Timothy EJ Behrens, Supervision, Funding acquisition, Project administration, Writing – review and editing; Erie Boorman, Conceptualization, Data curation, Formal analysis, Supervision, Funding acquisition, Investigation, Writing – original draft, Project administration

### Author ORCIDs

Phillip P Witkowski ⓘ https://orcid.org/0000-0001-7377-1592
Lindsay JH Rondot ⓘ https://orcid.org/0009-0007-6692-3237
Raymond J Dolan ⓘ https://orcid.org/0000-0001-9356-761X
Timothy EJ Behrens ⓘ https://orcid.org/0000-0003-0048-1177
Erie Boorman ⓘ https://orcid.org/0000-0002-8438-4499

Reviewer #1 (Public review): https://doi.org/10.7554/eLife.101841.3.sa1

Reviewer #2 (Public review): https://doi.org/10.7554/eLife.101841.3.sa2
Reviewer #3 (Public review): https://doi.org/10.7554/eLife.101841.3.sa3
Author response https://doi.org/10.7554/eLife.101841.3.sa4

## Additional files

### Supplementary files
MDAR checklist

### Data availability
Unthresholded group-level statistical maps have been deposited at NeuroVault (https://neurovault.org/collections/17702/) and are publicly available as of the date of publication. Links are listed in the key resources table. All original code has been deposited at Open Science Framework (https://osf.io/b9m6q/) and is publicly available as of the date of publication.

The following datasets were generated:

| Author(s) | Year | Dataset title | Dataset URL | Database and Identifier |
|---|---|---|---|---|
| Witkowski PP, Rondot LJH | 2024 | Neural mechanisms of credit assignment for delayed outcomes during associative learning | https://neurovault.org/collections/17702/ | NeuroVault, 17702 |
| Witkowski PP, Rondot LJH | 2024 | Neural mechanisms of credit assignment for delayed outcomes during associative learning | https://osf.io/b9m6q/ | Open Science Framework, b9m6q |

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
