## [Editor Report · eLife Assessment]

This study provides **important** findings that during credit assignment, the lateral orbitofrontal cortex (lOFC) and hippocampus (HC) encode causal choice representations, while the frontopolar cortex (FPl) mediates HC -lOFC interactions when the causality needs to be maintained over longer distractions. This research offers **compelling** evidence and employs sophisticated multivariate pattern analysis. However, while the task design captures the delayed component, it lacks the full complexity and ambiguity of the credit assignment process observed in real-world scenarios. Moreover, the data indicated that other frontal regions beyond just lOFC were involved in delayed credit assignment. This work will be of interest to cognitive and computational neuroscientists who work on value-based decision-making and fronto-hippocampal circuits.

---

## [Referee Report · Reviewer #1 (Public review)]

Summary

The authors conducted a study on one of the fundamental research topics in neuroscience: neural mechanisms of credit assignment. Building on the original studies of Walton and his colleagues and subsequent studies on the same topic, the authors extended the research into the delayed credit assignment problem with clever task design, which compared the non-delayed (direct) and delayed (indirect) credit assignment processes. Their primary goal was to elucidate the neural basis of these processes in humans, advancing our understanding beyond previous studies.

Major Strengths and Considerations

Strengths:

(1) Innovative task design distinguishing between direct and indirect credit assignment.

(2) Use of sophisticated multivariate pattern analysis to identify neural correlates of pending representations.

(3) Well-executed study with clear presentation of results.

(4) Extension of previous research to human subjects, providing valuable comparative insights.

Considerations for Future Research:

(1) The task design, while clear and effective, might be further developed to capture more real-world complexity in credit assignment.

(2) There's potential for deeper exploration of the role of task structure understanding in credit assignment processes.

(3) The interpretation of lateral orbitofrontal cortex (lOFC) involvement could be expanded to consider its role in both credit assignment and task structure representation.

Achievement of Aims and Support of Conclusions

The authors successfully achieved their aim of investigating direct and indirect credit assignment processes in humans. Their results provide valuable insights into the neural representations involved in these processes. The study's conclusions are generally well-supported by the data, particularly in identifying neural correlates of pending representations crucial for delayed credit assignment.

Impact on the Field and Utility of Methods

This study makes a significant contribution to the field of credit assignment research by bridging animal and human studies. The methods, particularly the multivariate pattern analysis approach, provide a robust template for future investigations in this area. The data generated offers valuable insights for researchers comparing human and animal models of credit assignment, as well as those studying the neural basis of decision-making and learning.

The study's focus on the lOFC and its role in credit assignment adds to our understanding of this brain region's function

Additional Context and Future Directions

(1) Temporal ambiguity in credit assignment: While the current design provides clear task conditions, future studies could explore more ambiguous scenarios to further reflect real-world complexity.

(2) Role of task structure understanding: The difference in task comprehension between human subjects in this study and animal subjects in previous studies offers an interesting point of comparison.

(3) The authors used a sophisticated method of multivariate pattern analysis to find the neural correlate of the pending representation of the previous choice, which will be used for credit assignment process in the later trials. The authors tend to use expressions that these representations are maintained throughout this intervening period. However the analysis period is specifically at the feedback period, which is irrelevant for the credit assignment of the immediately preceding choice. This task period can interfere with the interference of ongoing credit assignment process. Thus, rather than the passive process of maintaining the information of the previous choice, the activity of this specific period can mean the active process of protecting the information from interfering and irrelevant information. It would be great if the authors could comment on this important interpretational issue.

(4) Broader neural involvement: While the focus on specific regions of interest (ROIs) provided clear results, future studies could benefit from a whole-brain analysis approach to provide a more comprehensive understanding of the neural networks involved in credit assignment.

Comments after the revision:

The authors have adequately addressed the majority of concerns raised in my previous review. The manuscript has demonstrably improved as a result of these revisions and represents a valuable contribution to the literature on credit assignment.

However, some limitations persist that, while not readily resolvable within the scope of the current study, warrant attention. Specifically, the investigation focuses primarily on the temporal dimension of credit assignment. In real-world scenarios, the complexity of credit assignment extends beyond temporal distance to encompass the inherent ambiguity of causal attribution arising from the presence of multiple potential causal events. Resolving this ambiguity necessitates a form of structural understanding of the environment, a capacity presumably possessed by humans and animals. While the experimental design of this study provides explicit cues regarding the structure of the environment, deciphering such structure in natural settings is a crucial component of the credit assignment process.

Future research should prioritize the investigation of credit assignment within more ecologically valid contexts, focusing on the role of structural understanding in navigating the causal ambiguity inherent in real-world environments. Addressing this aspect will be crucial for developing a more complete and nuanced understanding of credit assignment mechanisms.

In addition, the newly added whole-brain searchlight decoding analysis provides an important nuance regarding the neural substrates of credit assignment (Figure S7). The results reveal not only activity in the lateral orbitofrontal cortex (lOFC), but also, and more robustly, in the medial orbitofrontal cortex/ventromedial prefrontal cortex (mOFC/vmPFC) specifically during the "indirect transition condition" and not the "direct transition condition." This finding suggests a potentially more significant role for mOFC/vmPFC in processing complex, non-immediate credit assignment scenarios. This nuance should be explicitly noted to appreciate the complexity of the neural mechanisms at play.

---

## [Referee Report · Reviewer #2 (Public review)]

Summary:

The present manuscript addresses a longstanding challenge in neuroscience: how the brain assigns credit for delayed outcomes, especially in real-world learning scenarios where decisions and outcomes are separated by time. The authors focus on the lateral orbitofrontal cortex and hippocampus, key regions involved in contingent learning. By integrating fMRI data and behavioral tasks, the authors examined how neural circuits maintain a causal link between past decisions and delayed outcomes. Their findings offer insights into mechanisms that could have critical implications for understanding human decision-making.

Strengths:

- The experimental designs were extremely well thought-out. The authors successfully coupled behavioral data and neural measures (through fMRI) to explore the neural mechanisms of contingent learning. This integration adds robustness to the findings and strengthens their relevance.

- The emphasis on the interaction between the lateral orbitofrontal cortex (lOFC) and hippocampus (HC) in this study is very well-targeted. The reported findings regarding their dynamic interactions provide valuable insights into contingent learning in humans.

- The use of advanced modeling framework and analytical techniques allowed the authors to uncover new mechanistic insights regarding a complex case of decision-making process. The methods developed will also benefit analyses of future neuroimaging data on a range of decision-making tasks as well.

Weaknesses:

- Given the limited temporal resolution of fMRI and that the measured signal is an indirect measure of neural activity, it is unclear the extent to which the reported causality reflects the true relationship/interactions between neurons in different regions. That said, I believe this concern is minimized by a series of well-thought-out and robust analyses which consistently point to compelling results.

Comments on revisions:

Thank you for your thorough point-by-point responses to my comments and questions. After carefully reviewing the responses and additional analyses/results provided, I do not have further comments. Importantly, I believe the authors have done a great job addressing inevitable limitations that are inherent to fMRI signals. The thoughtful analyses used in the study combined with the timely questions the manuscript is able to address make the study an important contribution to the field.

---

## [Referee Report · Reviewer #3 (Public review)]

The authors apply multivoxel decoding analyses from fMRI during reward feedback about the cues previously chosen that led to that feedback. They compare two versions of the task - one in which the feedback is provided about the current trial, and one in which the feedback is provided about the previous trial. Reward probability changes slowly over time, so subjects need to identify which cues are leading to reward at a given time. They find that evidence for recall of the cue in lateral orbitofrontal cortex (lOFC) and hippocampus (HC). They also find that in the second condition, where feedback is for the one-back trial, this representation is mediated by the lateral frontal pole (FPl).

Overall, the analyses are clean and elegant and seem to be complete. I have only a few comments, all of which can be public.

(1) They do find (not surprisingly) that the one-back task is harder. It would be good to ensure that the reason that they had more trouble detecting direct HC & lOFC effects on the harder task was not because the task is harder and thus that there are more learning failures on the harder one-back task. (I suspect their explanation that it is mediated by FPl is likely to be correct. But it would be nice to do some subsampling of the zero-back task [matched to the success rate of the one-back task] to ensure that they still see the direct HC and lOFC there.)

(2) The evidence that they present in the main text (Figure 3) that the HC and lOFC are mediated by FPl is a correlation. I found the evidence presented in Supplemental Figure 7 to be much more convincing. As I understand it, what they are showing in SF7 is that when FPl decodes the cue, then (and only then) HC and lOFC decode the cue. If my understanding is correct, then this is a much cleaner explanation for what is going on than the secondary correlation analysis. If my understanding here is incorrect, then they should provide a better explanation of what is going on so as to not confuse the reader.

(3) I like the idea of "credit spreading" across trials (Figure 1E). I think that credit spreading in each direction (into the past [lower left] and into the future [upper right]) is not equivalent. This can be seen in Figure 1D, where the two tasks show credit spreading differently. I think a lot more could be studied here. Does credit spreading in each of these directions decode in interesting ways in different places in the brain?

Comments on revisions:

After revision, I have no additional comments.

---

## [Author Response]

The following is the authors’ response to the original reviews.

**Reviewer 1:**
Point 1 of public reviews and point 2 of recommendations to authors.Temporal ambiguity in credit assignment: While the current design provides clear task conditions, future studies could explore more ambiguous scenarios to further reflect real-world complexity…. The role of ambiguity is very important for the credit assignment process. However, in the current task design, the instruction of the task design almost eliminates the ambiguity of which the trial's choice should be assigned credit to. The authors claim the realworld complexity of credit assignment in this task design. However, the real-world complexity of this type of temporal credit assignment involves this type of temporal ambiguity of responsibility as causal events. I am curious about the consequence of increasing the complexity of the credit assignment process, which is closer to the complexity in the real world.

We agree that the structure of causal relationships can be more ambiguous in real-world contexts. However, we also believe that there are multiple ways in which a task might approach “real-world complexity”. One way is by increasing the ambiguity in the relationships between choices and outcomes (as done by Jocham et al., 2016). Another is by adding interim decisions that must be completed between viewing the outcome of a first choice, which mimics task structures such as the cooking tasks described in the introduction. In such tasks, the temporal structure of the actions maybe irrelevant, but the relationship between choice identities and the actions is critical to be effective in the task (e.g., it doesn’t matter whether I add spice before or after the salt, all I need to know that adding spice will result in spicy soup). While ambiguity about either form of causal relation is clearly an important part of real-world complexity, and would make credit assignment harder, our study focuses on how links between outcomes and specific past choice identities are created at the neural level when they are known to be causal.

We consequently felt it necessary to resolve temporal ambiguity for participants. Instructing participants on the structure of the task allowed us to make assumptions about how credit assignment for choice identities should proceed (assign credit to the choice made N trials back) and allowed us make positive predictions about the content of representations in OFC when viewing an outcome. This gave the highest power to detect multivariate information about the causal choice and the highest interpretability of such findings.

In contrast, if we had not resolved this ambiguity, it would be difficult to tell if incorrect decoding from the classifier resulted from noise in the neural signal, or if on that trial participants were assigning credit to non-causal choices that they erroneously believed to have caused the outcome due to the perceived temporal structure. We believe this would have ultimately decreased our power to determine whether representations of the causal choice were present at the time of outcome because we would have to make assumptions about what counts as a “true” causal representation.

We have commented on this in the discussions (p.13):

“While our study was designed to focus on the complexity of assigning credit in tasks with different known causal structures, another important component of real-world credit assignment is temporal ambiguity. To isolate the mechanisms which create associations between specific choices and specific outcomes, we instructed participants on the causal structure of each task, removing temporal ambiguity about the causal choice. However, our results are largely congruent with previously reported results in tasks that dissolved the typical experimental trial structure, producing temporal ambiguity, and which observed more pronounced spreading of effect, in addition to appropriate credit assignment (Jocham et al, 2016). Namely, this study found that activation in the lOFC increased only when participants received rewards contingent on a previous action, an effect that was more pronounced in subjects whose behavior reflected more accurate credit assignment. This suggests a shared lOFC mechanism for credit assignment in different types of complex environments. Whether these mechanisms extend to situations where the temporal causal structure is completely unknown remains an important question.”

Point 2 of public reviews and point 1 of recommendations to authorsRole of task structure understanding: The difference in task comprehension between human subjects in this study and animal subjects in previous studies offers an interesting point of comparison…. The credit assignment involves the resolution of the ambiguity in which the causal responsibility of an outcome event is assigned to one of the preceding events. In the original study of Walton and his colleagues, the monkey subjects could not be instructed on the task structure defining the causal relationships of the events. Then, the authors of the original study observed the spreading of the credit assignments to the "irrelevant" events, which did not occur in the same trial of the outcome event but to the events (choices) in neighbouring trials. This aberrant pattern of the credit assignment can be due to the malfunctions of the credit assignment per se or the general confusion of the task structure on the part of the monkey subjects. In the current study design, the subjects are humans and they are not confused about the task structure. Consistently, it is well known that human subjects rarely show the same patterns of the "spreading of credit assignment". So the implicit mechanism of the credit assignment process involves the understanding of the task structure. In the current study, there are clearly demarked task conditions that almost resolve the ambiguity inherent in the credit assignment process. Yet, the focus of the current analysis stops short of elucidating the role of understanding the task structure. It would be great if the authors could comment on the general difference in the process between the conditions, whether it is behavioral or neural.

We would like to thank the reviewer for making this important point. We believe that understanding the structure of the credit-assignment problem above is quite important, at least for the type of credit assignment described here. That is, because participants know that the outcome viewed is caused by the choice they made, 0 or 1 trials into the past, they can flexibly link choice identities to the newly observed outcomes as the probabilities change. Note, however, that this is already very challenging in the 1-back condition because participants need to track the two independently changing probabilities. We believe this is critical to address the questions we aimed to answer with this experiment, as described above.

We agree that this might be quite different from previous studies done with non-human primates, which also included many more training trials and lesions to the lOFC. Both of these aspects could manifest as difference in task performance and processing at behavioural and neural levels, respectively. Consistent with this possibility, in our task, we found no differences in credit spreading between conditions, suggesting that humans were quite precise in both, despite causal relationships being harder to track in the “indirect transition condition”. This lack of credit spreading could be because humans better understood the task-structure compared to macaques or be due to differences in functioning of the OFC and other regions. Because all participants were trained to understand, and were cued with explicit knowledge of, the task structure, it is difficult to isolate its role as we would need another condition in which they were not instructed about the task structure. This would also be an interesting study, and we leave it to future research to parse the contributions of task-structure ambiguity to credit assignment.

Point 3 of public reviews.The authors used a sophisticated method of multivariate pattern analysis to find the neural correlate of the pending representation of the previous choice, which will be used for the credit assignment process in the later trials. The authors tend to use expressions that these representations are maintained throughout this intervening period. However, the analysis period is specifically at the feedback period, which is irrelevant to the credit assignment of the immediately preceding choice. This task period can interfere with the ongoing credit assignment process. Thus, rather than the passive process of maintaining the information of the previous choice, the activity of this specific period can mean the active process of protecting the information from interfering and irrelevant information. It would be great if the authors could comment on this important interpretational issue.

We agree that lFPC is likely actively protecting the pending choice representation from interference with the most recent choice for future credit assignment. This interpretation is largely congruent with the idea of “prospective memory” (e.g., Burgess, Gonen-Yaacovi, Volle, 2011), in which the lFPC can be thought of as protecting information that will be needed in the future but is not currently needed for ongoing behavior. That said, from our study alone it is difficult to make claims about whether the information maintained in frontal pole is actively protecting this information because of potentially interfering processes. Our “indirect transition condition” only contains trials where there is incoming, potentially interfering information about new outcomes, but no trials that might avoid interference (e.g., an interim choice made but there is nothing to be learned from it). We comment on this important future direction on page 14:

“One interpretation of these results is that the lFPC actively protects information about causal choices when potentially interfering information must be processed. Future studies will be needed to determine if the lFPC’s contributions are specific to these instances of potential interference, and whether this is a passive or active process”

Point 3 of recommendation to authorsA slightly minor, but still important issue is the interpretation of the role of lOFC. The authors compared the observed patterns of the credit assignment to the ideal patterns of credit assignment. Then, the similarity between these two matrices is used to find the associated brain region. In the assumption that lOFC is involved in the optimal credit assignment, the result seems reasonable. But as mentioned above, the current design involves the heavy role of understanding the task structure, it is debatable whether the lOFC is just involved in the credit assignment process or a more general role of representing the task structure.

We agree that this is an important distinction to make, and it is very likely that multiple regions of the OFC carry information about the task structure, and the extent to which participants understood this structure may be reflected in behavioral estimates of credit assignment or the overall patterns of the matrices (though all participants verbalized the correct structure prior to the task). However, we believe that in our task the lOFC is specifically involved in credit-assignment because of the content of the information we decoded. We demonstrated that the lOFC and HPC carry information about the causal choice during the outcome. These results cannot be explained by differences in understanding of the task structure because that understanding would have been consistent across trials where participants choose either shape identity. Thus, a classifier could not use this to separate these types of trials and would reflect chance decoding.

One interpretation of the lOFC’s role in credit assignment is that it is particularly important when a model of the task structure has to be used to assign credit appropriately. Here, we show lOFC the reinstates specific causal representations precisely at the time credit needs to be assigned, which are appropriate to participants’ knowledge of the task structure. These representations may exist alongside representations of the task structure, in the lOFC and other regions of the brain (Park et al., 2020; Boorman et al., 2021; Seo and Lee, 2010; Schuck et al., 2016). We have added the following sentences to clarify our perspective on this point in the discussion (p. 13):

“Our results from the “indirect transition” condition show that these patterns are not merely representations of the most recent choice but are representations of the *causal* choice given the current task structure, and may exist alongside representations of the task structure, in the lOFC and elsewhere (Boorman et al., 2021; Park et al., 2020; Schuck et al., 2016; Seo & Lee, 2010).”

Point 4 of public reviews and point 4 of recommendation to authorsBroader neural involvement: While the focus on specific regions of interest (ROIs) provided clear results, future studies could benefit from a whole-brain analysis approach to provide a more comprehensive understanding of the neural networks involved in credit assignment… Also, given the ROI constraint of the analysis, the other neural structure may be involved in representing the task structure but not detected in the current analysis

Given our strong a priori hypotheses about regions of interest (ROIs) in this study, we focused on these specific areas. This choice was based on theoretical and empirical grounds that guided our investigation. However, we thank the reviewer for pointing this out and agree that there could be other unexplored areas that are critical to credit-assignment which we did not examine.

We conducted the same searchlight decoding procedure on a whole brain map and corrected for multiple comparisons using TFCE. We found no significant regions of the brain in the “direct transition condition” but did find other significant regions in our information connectivity analysis of the “indirect transition condition”. In addition to replicating the effects in lOFC and HPC, we also found a region of mOFC which showed a strong correlation with pending choice in lFPC. It’s difficult to say whether this region is involved in credit assignment *per se*, because we did not see this region in the “direct transition condition” and so we cannot say that it is consistently related to this process. However, the mOFC is thought to be critical to representing the current task state (Schuck et al., 2016), and the task structure (Park et al., 2020). In our task, it could be a critical region for communicating *how* to assign credit given the more complex task structure of the “indirect transition condition” but more evidence would be needed to support this interpretation.

For now, we have added the results of this whole brain analysis to a new supplementary figure S7 (page 41), and all unthresholded maps have been deposited in a Neurovault repository, which is linked in the paper, for interested readers to assess.

Minor points:There are some missing and confusing details in the Figure reference in the main text. For example, references to Figure 3 are almost missing in the section "Pending item representations in FPl during indirect transitions predict credit assignment in lOFC". For readability, the authors should improve this point in this section and other sections.

Thank you to the reviewer for pointing this out. We have now added references to Figure 3 on page 8:

“Our analysis revealed a cluster of voxels specifically within the right lFPC ([x,y,z] = [28, 54, 8], *t*(19) = 3.74, *pTFCE* <0.05 ROI-corrected; left hemisphere all *pTFCE* > 0.1, Fig. 3A)”

And on page 10:

Specifically, we found significant correlations in decoding distance between lFPC and bilateral lOFC ([x,y,z] = [-32,24, -22], *t*(19) = 3.81, [x,y,z] = [20, 38, -14], *t*(19) = 3.87, *pTFCE* <0.05 ROI corrected) and bilateral HC ([x,y,z] = [-28, -10, -24], *t*(19) = 3.41, [x,y,z] = [22, -10, -24], *t*(19) = 4.21, *pTFCE* <0.05 ROI corrected), Fig. 3C.

Task instructions for the two conditions (direct and indirect) play important roles in the study. If possible, please include the following parts in the figures and descriptions in the introduction and/or results sections.

We have now included a short description of the condition instructions beginning on page 5:

“Participants were instructed about which condition they were in with a screen displaying “Your latest choice” in the direct transition condition, and “Your previous choice” in the indirect condition.”

And have modified Figure 1 to include the instructions in the title of each condition. We thought this to be the most parsimonious solution so that the choice options in the examples were not occluded.

The subject sample size might be slightly too small in the current standards. Please give some justifications.

We originally selected the sample size for this study to be commensurate with previous studies that looked for similar behavioral and neural effects (see Boorman et al., 2016; Howard et al., 2015; Jocham et al., 2016). This has been mentioned in the “methods” section on page 24.

However, to be thorough, we performed a power analysis of this sample size using simulations based on an independently collected, unpublished data set. In this data set, 28 participants competed an associative learning task similar to the task in the current manuscript. We trained a classifier to decode causal choice option at the time of feedback, using the same searchlight and cross-validation procedures described in the current manuscript, for the same lateral OFC ROI. We calculated power for various sample sizes by drawing N participants with replacement 1000 times, for values of N ranging from 15 to 25. After sampling the participants, we tested for significant decoding for the causal choice within the subset of data, using smallvolume TFCE correction to correct for multiple comparisons. Finally, we calculated the proportion of these samples that were significant at a level of *pTFCE* <.05.

The results of this procedure show that an N of 20 would result in 84.2% power, which is slightly above the typically acceptable level of 80%. We have added the following sentences to the methods section on page 25:

“Using an independent, unpublished data set, we conducted a power analysis for the desire neural effect in lOFC. We found that this number of participants had 84% power to detect this effect (Fig. S8).”

We also added the following figure to the supplemental figures page (42):

**Reviewer 2:**
I have several concerns regarding the causality analyses in this study. While Multivariate analyses of information connectivity between regions are interesting and appear rigorous, they make some assumptions about the nature of the input data. It is unclear if fMRI with its poor temporal resolution (in addition to possible region-specific heterogeneity in the readouts), can be coupled with these casual analysis methods to meaningfully study dynamics on a decision task where temporal dynamics is a core component (i.e., delay). It would be helpful to include more information/justification on the methods for inferring relationships across regions from fMRI data. Along this line, discussing the reported findings in light of these limitations would be essential.

We agree that fMRI is limited for capturing fast neural dynamics, and that it can be difficult to separate events that occur within a few seconds. However, we designed the information connectivity analysis to maximally separate the events in question – the representations of the causal choice being held in a pending state, and the representation of the causal choice during credit assignment. These events were separated by at least 10 seconds and by 15 seconds on average, which is commensurate with recommended intervals for disentangling information in such analysis (Mumford et al., 2012, 2014, also see van Loon et al., 2018, eLife; as example of fluctuations in decodability over time). This feature of our task design may not have been clear because information connectivity analyses are typically performed in the same task period. We clarify this point on page 32:

“Note that the decoding fidelity metric at each time point represents the decodability of the same choice at different phases of the task. These phases were separated by at least 10 seconds and 15 seconds on average, which can be sufficient for disentangling unique activity (Mumford et al., 2012, 2014).”

However, we agree with the reviewer that the limitations of fMRI make it difficult to precisely determine how roles of the OFC and lFPC might change over time, and whether other regions may contribute to information transfer at times scales which cannot be detected by fMRI. Further, we do not wish to imply causality between lFPC and lOFC (something we believe we do not claim in the paper), only that information strength in lFPC *predicts* subsequent strength of the same information in the OFC and HC. We have clarified this limitation on page 14:

“Although we show evidence that lFPC is involved in maintaining specific content about causal choices during interim choices, the limited temporal resolution of fMRI makes it difficult to tell if other regions may be supporting the learning processes at timescales not detectable in the BOLD response. Thus, it is possible that the network of regions supporting credit assignment in complex tasks may be much larger. Our results provide a critical first stem in discerning the nature of interactions between cognitive subsystems that make different contributions to the learning process in these complex tasks.”

**Reviewer 3:**
Point 1 of public reviews:They do find (not surprisingly) that the one-back task is harder. It would be good to ensure that the reason that they had more trouble detecting direct HC & lOFC effects on the harder task was not because the task is harder and thus that there are more learning failures on the harder oneback task. (I suspect their explanation that it is mediated by FPl is likely to be correct. But it would be nice to do some subsampling of the zero-back task [matched to the success rate of the one-back task] to ensure that they still see the direct HC and lOFC there).

We would like to thank the reviewer for this comment and agree that the “indirect transition condition” is more difficult than the direct transition condition. However, in this task it is difficult to have an explicit measure of learning failures *per se* because the “correctness” of a choice is to some extent subjective (i.e., based on the gift card preference and the computational model). We could infer when learning failures occur through the computational model by looking at trials in which participants made choices that the model would consider improbable, (i.e., non-reward maximizing) while accounting for outcome preference. However, there are also a myriad of other possible explanations for these choices, such as exploratory/confirmatory strategies, lapses in attention etc. Thus, we could not guarantee that the two conditions would be uniquely matched in difficulty with specific regard to learning even if we subsampled these trials. We feel it would be better left to future experiments which can specifically compare learning failures to tackle this issue. We have now addressed this point when discussing the model on page 31:

“Note that learning failures are not trivial to identify in our paradigm and model, because every choice is based on a participant’s preference between gift card outcomes, and the ability of the computational model to accurately estimate participants’ beliefs in the stimulus-outcome transition probabilities.”

Point 2 of public reviews:The evidence that they present in the main text (Figure 3) that the HC and lOFC are mediated by FPl is a correlation. I found the evidence presented in Supplemental Figure 7 to be much more convincing. As I understand it, what they are showing in SF7 is that when FPl decodes the cue, then (and only then) HC and lOFC decode the cue. If my understanding is correct, then this is a much cleaner explanation for what is going on than the secondary correlation analysis. If my understanding here is incorrect, then they should provide a better explanation of what is going on so as to not confuse the reader.

SF7 (now Figures 3C and 3D) does show that positive decoding in the HC and lOFC are more likely to occur when there is positive decoding in lFPC. However, the analysis shown in these figures are only meant to be control analysis to further characterise what is being captured, but not necessarily implied, by the information connectivity analysis. For example, in principle the classifier might never correctly decode a choice label in the lOFC or HC while still getting closer to the hyperplane when the lFPC patterns are correctly decoded. This would lead to a positive correlation, but a difficult to interpret result since patterns in lOFC and HPC are incorrect. Figure SF7A (now Fig. 3C) shows that this is not the case. Lateral OFC and HC have higher than chance positive decoding when lFPC has positive decoding. Figure SF7B (now Fig. 3D) shows that we can decode that information even if a new hyperplane is constructed. However, both cases have less information about the relationship between these regions because they do not include the trials where lOFC/HC and lFPC classifiers were incorrect at the same time. The correlation in Figure 3B includes these failures, giving a more wholistic picture of the data. We therefore try to concisely clarify this point on page 10:

“These signed distances allow us to relate both success in decoding information, as well as failures, between regions.”

And here on page 10:

“Subsequent analyses confirmed that this effect was due to these regions showing a significant increase in positive (correct) decoding in trials where pending information could be positively (correctly) decoded in lFPC, and not simply due to a reduction in incorrect information fidelity (see Fig. 3C & 3D).”

And have integrated these figures on page 9:

Point 3 of public reviews:

I like the idea of "credit spreading" across trials (Figure 1E). I think that credit spreading in each direction (into the past [lower left] and into the future [upper right]) is not equivalent. This can be seen in Figure 1D, where the two tasks show credit spreading differently. I think a lot more could be studied here. Does credit spreading in each of these directions decode in interesting ways in different places in the brain?

We agree that this an interesting question because each component of the off diagonal (upper and lower triangles) may reflect qualitatively different processes of credit spreading. However, we believe this analysis is difficult to carry out with the current dataset for two reasons. First, we designed this study to ask specifically about the information represented in key credit assignment regions during precise credit assignment, meaning we did not optimize the task to induce credit spreading at any point. Indeed, our efforts to train participants on the task were to ensure they would correctly assign credit as much as possible. Figure 1F shows that the regression coefficients representing credit spreading in each condition are near zero (in the negative direction), with little individual differences compared to the credit assignment coefficients. Thus, any analysis aiming to test for credit spreading would unfortunately be poorly powered. Studies such as Jocham et al. (2016), with more variability in causal structures, or studies with ambiguity about the causal structure by dissolving the typical trial structure would be better suited to address this interesting question. The second reason why such an analysis would be challenging is that due to our design, it is difficult to intuitively determine what kind of information should be coded by neural regions when credit spreads to the upper diagonal, since these cells reflect current outcomes that are being linked to future choices.

Replace all the FPl with LFPC (lateral frontal polar cortex)

We have no replace “FPl” with “LFPC” throughout the text and figures